# Picosecond-scale heterogeneous melting of metals at extreme non-equilibrium states

Qiyu Zeng [1,2,3], Xiaoxiang Yu [1,2,3] ✉, Bo Chen [1,2,3], Shen Zhang[1,2,3], Kaiguo Chen [1,2,3], Dongdong Kang [1,2,3] ✉ & Jiayu Dai [1,2,3] ✉

Extreme electron-ion non-equilibrium states, generated by ultrafast laser excitation, lead to melting processes that are fundamentally different from those under conventional thermal equilibrium and remain not fully understood. Through neural network-enhanced multiscale simulations of tungsten and gold nanofilms, we identify electronic pressure relaxation as critical to heterogeneous phase transformations. This nonthermal expansion generates a density decrease that enable surface-initiated melting far below equilibrium melting temperatures, creating electronic pressure-driven solid-liquid interface propagation at a high speed of 2500 ms$^{-1}$—tenfold faster than that of thermal heterogeneous melting mechanisms. Simulated time-resolved X-ray diffraction signatures distinguish this nonthermal expansion from thermal expansion dynamics driven by thermoelastic stress. These results establish hot-electron-mediated lattice destabilization as a universal pathway for laser-induced structural transformations, providing new insights for interpreting time-resolved experiments and controlling laser-matter interactions.

Ultrafast laser excitation has emerged as a transformative tool for probing and manipulating matter under extreme non-equilibrium conditions, enabling breakthroughs from attosecond spectroscopy to high-precision nanofabrication[1,2]. A hallmark of these interactions is the transient electronic excitation that establishes a pronounced temperature disparity between electrons and lattices ($T_e \gg T_i$)—a non-equilibrium regime where hot electrons coexist with a cold ionic framework. This state, which differs from thermal equilibrium ($T_e = T_i$), dynamically reconfigures interatomic interactions: charge density redistribution modulates potential energy surfaces (PESs), altering bonding forces and energy barriers[3,4]. Such processes deviate from equilibrium thermodynamic pathways, raising fundamental questions about how materials evolve when energy deposition outpaces thermal relaxation, particularly in laser-induced melting.

The isochoric hypothesis, which assumes negligible volume changes during sub-picosecond heating events, has been the cornerstone of our understanding of material response upon laser excitation. Within this picture, a plethora of femtosecond phenomena have been

uncovered: bond hardening[5,6], phonon softening[7], and dynamic lattice instabilities[8,9]. These processes occur on timescales of ~$10^2$ fs, where inertial stress confinement ensures a quasi-constant density. However, emerging evidence reveals a significant shift in understanding material response at picosecond timescales: hot electrons can exert an additional pressure component $p_e$—besides the thermoelastic stress—through thermal kinetic energy and quantum degeneracy of thermalized electrons[10–12], resulting in a volume change that may challenge the isochoric assumption. In gold, for instance, isobaric calculations predict phonon softening and nonthermal melting[13,14], in stark contrast with the isochoric hardening evidenced in sub-ps X-ray diffraction (XRD) experiments[15]. This divergence highlights a critical knowledge gap: How does the accumulation of electronic pressure (during ~$10^2$ fs) and its relaxation (on ~$10^1$ ps timescale) affect structural transformations?

Addressing this issue is challenging because the laser-driven process involves several coupled physical processes across different scales, including laser excitation that modifies the PES, non-adiabatic

[1]College of Science, National University of Defense Technology, Changsha, Hunan, China. [2]Hunan Key Laboratory of Extreme Matter and Applications, National University of Defense Technology, Changsha, Hunan, China. [3]Hunan Research Center of the Basic Discipline for Physical States, National University of Defense Technology, Changsha, China. ✉e-mail: xxyu@nudt.edu.cn; ddkang@nudt.edu.cn; jydai@nudt.edu.cn

electron-ion energy exchange, and atomic-scale structural responses[9,16]. Experimentally, the above issues, along with inherent limitations in temporal and spatial resolution, confound the interpretation of measured data[17–20]. Theoretically, the laser-driven process presents dual requirements that existing methodologies struggle to reconcile: ab initio accuracy to capture the hot-electron-modified PES governing $p_e$, and atomic-scale resolution across experimentally relevant thickness (tens of nanometers) to track stress wave propagation and phase transition. Ab initio accuracy and large-scale simulation size significantly constrain the efficacy of existing methodologies, such as classical molecular dynamics combined with two-temperature model[21,22] or real-time time-dependent density functional theory[23–25], thereby hindering a thorough exploration into the real-time response of laser-excited material without any prior constraints.

Here, we unify ab initio accuracy with large-scale molecular dynamics using recently developed hybrid atomistic-continuum approach, known as the two-temperature model coupled deep potential molecular dynamics (TTM-DPMD)[12]. By simulating free-standing tungsten (W) and gold (Au) nanofilms, we uncover the crucial role of electronic pressures in triggering heterogeneous structural transformations at picosecond scales. Time-resolved XRD signatures further distinguish this mechanism from thermal expansion, offering experimental validation pathways.

## Results

### Ab initio modeling of laser-driven dynamics
Within the TTM-DPMD framework, the electron-temperature-dependent deep neural network (ETD-DNN) is implemented to capture the ab initio laser-excited PES while maintaining computational efficiency. With ETD-NN, the nonthermal contribution to total energy, forces, and pressure at elevated electronic temperature $T_e$ is inherently incorporated (see "Methods" section for more details about ETD-NN). To ensure the accuracy of our neural network model, we have validated the thermophysical and vibrational properties of W and Au under both equilibrium ($T_e = T_i$) and non-equilibrium conditions ($T_e \neq T_i$)[12]. Especially, we reproduced the phonon softening in W along the H−N and N−Γ paths in the first Brillouin zone at elevated electron temperatures up to 20,000 K, consistent with previous predictions[26] (see Supplementary Fig. 1). Under a more severe non-equilibrium state ($T_e = 22,000$ K), an imaginary phonon mode at the N-point was observed, indicating a possible solid-solid phase transition. Here, we mainly focus on the moderate non-equilibrium condition below the $T_e = 20,000$ K to exclude the influence of lattice-instability-driven nonthermal phase transition.

The efficiency of TTM-DPMD approaches enables a full-scale ab initio description where the geometry of the sample is compatible with the experimental conditions. Here, we choose 30-nm-thick W nanofilm[27] and 35-nm-thick Au nanofilm[18] as the target sample. Since the foil thickness is comparable to the mean free path of excited electrons (~33 nm in W and ~100 nm in Au), the ballistic electron transport and reflux inside the foil produce uniform laser energy deposition[28]. Free boundary condition is applied to the laser incident direction ($z$-axis) to allow free surface response to the relaxation of internal stress (including both electronic and thermoelastic contributions). Periodic boundary conditions are applied in the lateral directions to simulate the experimental conditions where the laser spot diameter is large (hundreds of microns) compared to the depth of laser energy deposition[29]. More details about the TTM-DPMD simulation are provided in the "Methods" section.

### Ultrafast heterogeneous melting dynamics
Here we present the structural transformation of laser-excited W subjected to an absorbed laser fluence of 120 mJ cm$^{-2}$ (duration of laser pulse set to 130 fs). This fluence, although insufficient to trigger dynamical instability in the BCC lattice, reveals significant alterations in melting behavior due to the presence of hot electrons.

For comparison, we firstly discuss the purely thermal process obtained from a traditional TTM-MD method with ground-state PES that does not incorporate laser modification (Fig. 1a). After laser energy deposition, the system reaches maximum electron temperature $T_e = 19,050$ K while the lattice remains cold ($T_i = 300$ K). The lattice temperature quickly increases due to electron-phonon energy exchange. Once lattice temperature exceeds the limit of lattice thermal stability, the nucleation of a liquid region inside the foil is triggered, known as the homogeneous melting mechanism[21]. Within the first 5 ps, surface expansion remains limited to moderate volumetric change (Supplementary Fig. 2a), as constrained by the timescale of thermal pressure buildup.

The introduction of laser modification in PES through our ETD-NN model fundamentally alters this picture. As presented in Fig. 1b, the W nanofilm quickly responds to laser heating on a sub-picosecond timescale and exhibits instant surface expansion not observed in a purely thermal process. The heterogeneous nucleation of the FCC phase is initially formed in the surface region, then the BCC-FCC transformation front moves inward with an average velocity of 2571 m s$^{-1}$. Subsequently, as ion temperature increases above ≈2200 K at $t = 2$ ps, although the temperature is significantly lower than the melting point ($T_m = 3540$ K) under ambient conditions, the collapse of crystal structure occurs in the surface region and proceeds inward as the BCC-FCC interface propagates. Such a melting process exhibits a well-defined solid-liquid interface moving inward, which is similar to the heterogeneous melting mechanism but has a high propagation speed up to ≈2500 m s$^{-1}$. Here, we name this melting behavior as "ultrafast heterogeneous melting" because the melting speed is an order of magnitude greater than that of the conventional heterogeneous melting process (~10$^2$ m s$^{-1}$)[18]. After $t = 4$ ps, the melt front rapidly advances, completing the melting process within the following 2 ps.

### Electronic pressure-induced thermodynamic and structural heterogeneity
The ultrafast heterogeneous melting process exhibits spatial characteristics akin to conventional heterogeneous melting, with a well-defined melt front, while temporally resembling homogeneous melting due to its rapid progression. Such a unique combination of spatial and temporal features motivates a deeper exploration of the thermodynamic pathways and microscopic mechanisms involved.

As shown in Fig. 2, we present the thermodynamic evolution and microscopic structural transformation experienced by the interior ($z = 14.2$ nm) and surface ($z = 1.8$ nm) regions of the nanofilm. We observe that ultrafast electron heating induces extreme internal stress buildup within sub-picosecond timescales (Fig. 2a). The hot electrons can contribute to an extra stress of ≈45 GPa, which is significantly larger than that from thermal atomic vibrations of the lattice. With the existence of a free surface, a super-high expansion velocity of ≈755 m s$^{-1}$ at the initial state ($t = 0.4$ ps) is observed (see Supplementary Fig. 3), and the uniaxial expansion process launches instantly to release this hot-electron-induced pressure. As shown in Fig. 2b, the density in the surface region quickly decreases while the interior region is heated under isochoric conditions during the first 3 ps. As a consequence, the inhomogeneous thermodynamic profiles are created. In the density profile, a well-defined interface with a sharp decrease of $\Delta\rho \approx 5$ g cm$^{-3}$ follows the release of stress (Fig. 2b). This reduced density strongly influences the thermal stability of the lattice and explains the early onset of surface disordering below the equilibrium melting point.

From a microscopic perspective, the relaxation of electronic stress waves induces local structural transformations. The uniaxial expansion increases interatomic distances perpendicular to the free

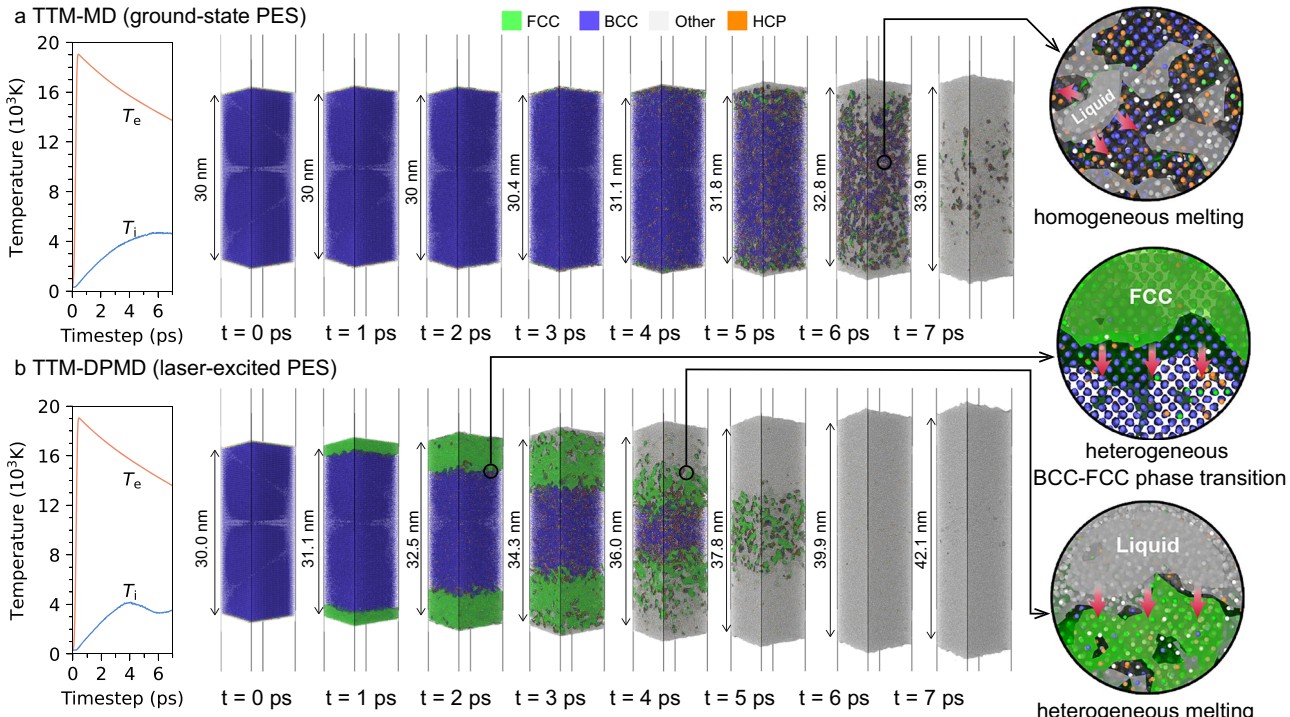

**Fig. 1 | Diverging melting dynamics in laser-excited tungsten.** Under absorbed laser fluence of 120 mJ cm⁻², **a** conventional TTM-MD prediction with ground-state PES $A(\mathcal{R})$ showing homogeneous melting governed by electron-phonon coupling. **b** Our TTM-DPMD results with laser-excited PES $A(\mathcal{R}, T_e)$ revealing electronic pressure-driven heterogeneous melting. The local structures are identified by the polyhedral template matching (PTM) method[48], and polyhedral surface meshes around FCC-type (green) and amorphous-type (gray) particles are constructed to highlight the heterogeneity in lattice symmetry. Red arrows indicate the propagation direction of phase transition (solid-liquid or BCC-FCC) interfaces. The atomic configurations are visualized by OVITO software[49].

surface, leading to a transformation from a distorted BCC structure to an FCC structure along the uniaxial Bain deformation path[30]. This observation is similar to the BCC-FCC transition observed by Murphy et al. using isotropic isobaric ab initio MD[31], highlighting the unique role of electronic pressure in modifying phase transitions. As shown in Fig. 2c–f, the short-range order in the laser-excited nanofilm confirms the coexistence of high-density BCC and low-density FCC structures. At $t = 1.5$ ps, the interior atoms display characteristics of a thermally fluctuated BCC structure, while the surface atoms present typical FCC close-packed structures. This depth-variation in lattice coordination suggests that the density discontinuity driven by electronic pressure is accompanied by a disruption in local lattice symmetry. As the temperature continues to rise, the interplay between lattice heating and electronic pressure relaxation creates a multi-phase system composed of high-density BCC, low-density FCC, and disordered structures (Supplementary Fig. 4).

## Discussion

### Laser fluence dependence of structural transformation

To comprehensively understand the lattice response driven by electronic pressure, we performed molecular dynamics simulations under an uniaxial isobaric-isothermal ensemble ($T_i = 300$ K, $p = 1$ bar) across a wide range of $T_e$ from 10,000 K to 20,000 K.

The equilibrated density and lattice structure are shown in Fig. 3a. The structural transformation dynamics exhibits obvious dependence on laser fluence. Under a relatively moderate two-temperature state ($T_e \leq 13,000$ K), the BCC structure is maintained with slight uniaxial distortion. The density decreases from an initial value of 19.15 g cm⁻³ to 17.15 g cm⁻³ at $T_e = 13,000$ K, with a uniaxial strain of ≈8.81%. As the $T_e$ increases, the electronic pressure drives the initial BCC structure towards the more close-packed structure, accompanied by shuffling of several close-packed atomic planes, thus obtaining the FCC structure with stacking faults. When the $T_e$ exceeds 16,000 K, the stacking faults

disappear and a pure FCC structure is generated with a density of 14.13 g cm⁻³ at $T_e = 20,000$ K, where the corresponding uniaxial strain is as high as 35.43%. A series of TTM-DPMD simulations under different laser fluence confirmed the BCC-FCC phase transition threshold (Supplementary Fig. 5). Once the laser fluence exceeds 80 mJ cm⁻², corresponding to a maximum electron temperature of 15,000 K, the heterogeneous nucleation and growth of FCC structure begin to appear as the uniaxial expansion proceeds.

Further, we investigate the laser fluence dependence of the melting mechanism. By calculating the ultrafast electron diffraction pattern, we determine the complete melting time through the decay of (110) diffraction peak during laser heating processes (Supplementary Fig. 6). For comparison, the results from purely thermal simulation are discussed (Supplementary Fig. 7). As the laser fluence increases near the melting threshold (53 mJ cm⁻²), the thermal heterogeneous melting starts from the free surface and proceeds slowly by the subsonic melt-front propagation (≈125 m s⁻¹), such process lasts hundreds of picoseconds. At high laser fluence, the homogeneous melting dominates, where the nucleation and growth of the liquid region inside the foil quickly completes the melting within several picoseconds.

When incorporating the laser-excited PES through our ETD-NN model, the threshold fluences to the different melting regimes are reduced; both heterogeneous and homogeneous melting are predicted to occur more rapidly. Moreover, as shown in Fig. 3b, the release of electronic pressure introduces a sharp decrease in local density, resulting in a significantly reduced melting point in the surface region as compared with the isochoric condition. Therefore, as ion temperature increases, the low-density surface crystalline can quickly collapse into a disordered structure before the interior melting. Based on TTM-DPMD simulations (Supplementary Fig. 5), here we can specify the laser fluence needed for triggering the newly discovered "ultrafast heterogeneous melting" process in Fig. 3c. As the initial electron temperature exceeds $T_e = 16,000$ K, the isobaric melting point can

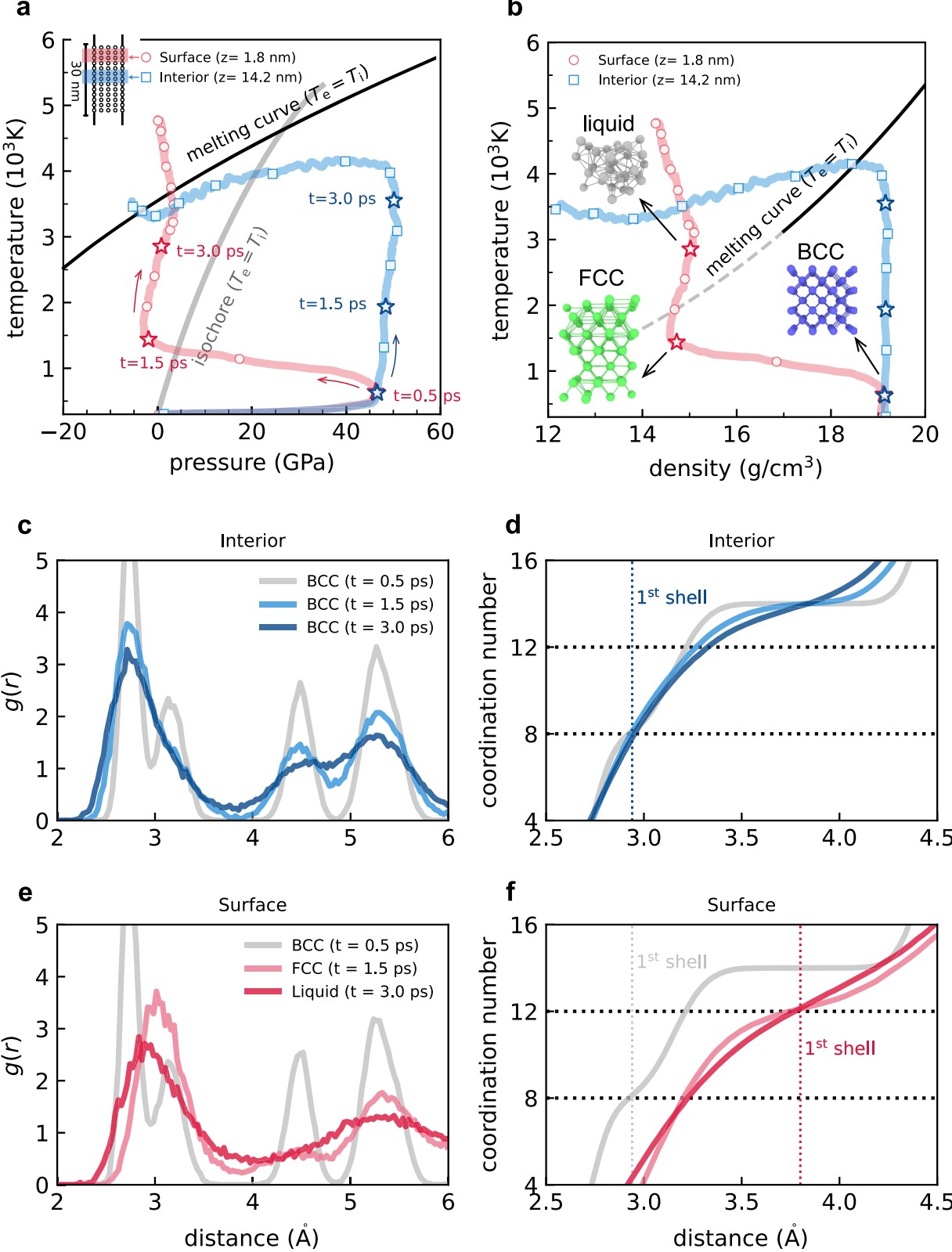

**Fig. 2 | Extreme heterogeneity in laser-excited W. a, b** Depth-dependent thermodynamic pathways in tungsten nanofilm under laser fluence of 120 mJ cm$^{-2}$. Colored circles show thermodynamic states at 0.5 ps intervals, with arrows indicating evolution trajectories. Red stars mark three representative thermodynamic states of the surface region ($z = 1.8$ nm) corresponding to: initial electronic pressure buildup ($t = 0.5$ ps), complete pressure release ($t = 1.5$ ps), and surface melting ($t = 3$ ps). Blue stars denote simultaneous thermodynamic states of the interior region ($z = 14.2$ nm) for comparison. The calculated equilibrium isochore (gray solid line) and melting curve (black solid line) are also presented for comparison[12]. **c, e** Radial distribution functions $g(r)$ and **d, f** coordination numbers (CN) for surface and interior regions at selected times, where the radial distribution of CN is obtained by integration of $g(r)$.

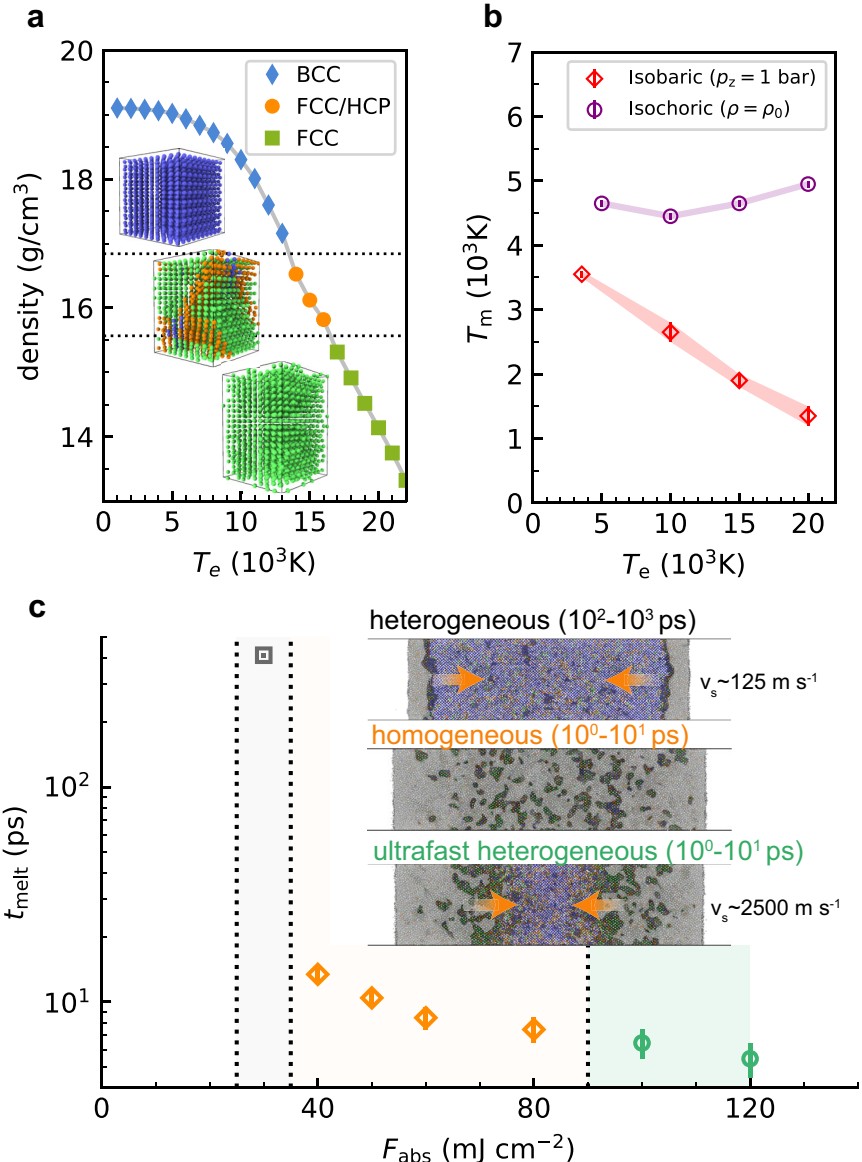

**Fig. 3 | Laser fluence dependence of structural transformation in W. a** Density decrease after electronic pressure relaxation along (100) direction, insets denote the uniaxially-distorted BCC, FCC with stacking faults, and FCC, respectively. **b** Isochoric and isobaric melting behavior under non-equilibrium conditions, obtained from fixed-$T_e$ DPMD simulations via the two-phase method. The error bar associated with the melting point is defined as half the temperature interval between the two-phase method simulations, where the solid phase is stable and those where it is molten. **c** Complete melting time under different laser fluence. The yellow, red, and green regions denote the heterogeneous, homogeneous, and ultrafast heterogeneous melting mechanisms, respectively. The error bar for the complete melting time is defined as the temporal resolution of the atomic trajectory output from the TTM-DPMD simulations.

deviate as large as $\Delta T_m \approx 2750$ K from the isochoric one. Under such conditions, the ultrafast heterogeneous process dominates the melting dynamics.

**Nonthermal expansion signatures in X-ray diffraction lineouts**
To elucidate the unique expansion dynamics under electronic pressure, we analyze the static structure factor $S(q)$, which directly correlates with XRD measurements. During the laser heating process, internal stress is released at the surface, generating stress waves that propagate inward. This results in a mixed signal representing both the expanded surface region and the isochoric interior region, as captured by $S(q)$.

As shown in Fig. 4, we take laser-excited W with laser fluence of $F_{abs} = 80$ mJ cm$^{-2}$ as an example. In the purely thermal process, the dominant role of electron-phonon coupling leads to the accumulation of thermal kinetic pressure inside the foil, thus driving gradual surface expansion. From Fig. 4a, the new characteristic peaks split from the

original peak at $q = 2.8$ Å$^{-1}$, then continuously shift to lower $q$ values, corresponding to the gradual increase in interatomic distance of the surface region under uniaxial expansion.

In nonthermal processes (Fig. 4b), the generation of electronic pressure originates from the formation of the initial two-temperature state. Consequently, the release of hot-electron-induced pressure occurs at sub-picosecond timescales, resulting in significant uniaxial distortion. At $t = 2$ ps, the proportion of low-density surface region reaches a detectable level for XRD, resulting in a notable splitting of the first diffraction peak. A new characteristic peak has been observed at $q_s = 2.613$ Å$^{-1}$. Unlike thermal processes, the position of this new peak, $q_s$, remains unchanged over time. Only its intensity gradually increases, indicating the ongoing nonthermal expansion process. As the lattice temperature increases, the long-range structure of the lattice progressively dissipates, leading to the disappearance of characteristic peaks associated with crystalline structure in $S(q)$ at $t = 7$ ps.

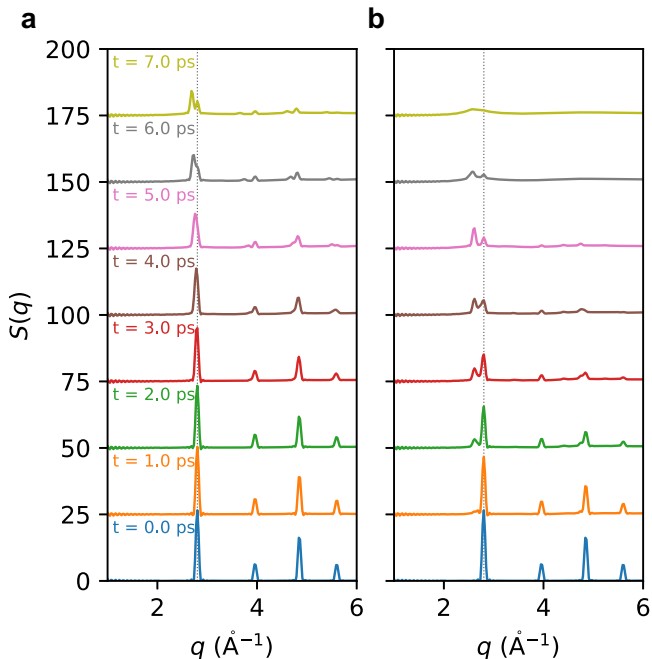

**Fig. 4 | Distinct thermal versus nonthermal expansion behavior revealed by X-ray diffraction.** Static structure factor $S(q)$ of laser-excited W nanofilm under laser fluence of 80 mJ cm$^{-2}$, estimated from TTM-DPMD trajectories with **a** ground-state PES and **b** laser-excited PES. The black line highlights the first diffraction peak from the initial BCC structure.

As shown in Fig. 5, we present the temporal evolution of $S(q)$ under various laser fluences, ranging from 60 mJ cm$^{-2}$ to 120 mJ cm$^{-2}$, aiming to provide a comprehensive understanding of expansion dynamics. Analysis of the 2.4–3.0 Å$^{-1}$ diffraction regime reveals distinct fluence-dependent behaviors: (1) During thermal expansion, it manifests as continuous shifting, where a higher lattice heating rate can result in a slightly faster shift of the new diffraction peak, as shown in Fig. 5a. (2) During the nonthermal expansion process, it exhibits discontinuous splitting, and the position of the new diffraction peak displays a strong dependence on laser fluence. A larger electronic pressure corresponds to a lower $q$ value position of the new peak, as illustrated in Fig. 5b. The observed qualitative disparity in expansion dynamics, as indicated by $S(q)$, implies the possibility of identifying evidence of electronic pressure relaxation dynamics through time-resolved XRD experimental methodologies.

### Influence of electron-phonon coupling strength and sample thickness

We further examined the effects of electron-phonon coupling strength $G(T_e)$ and sample thickness on the ultrafast heterogeneous melting behavior. Although theoretical predictions for $G(T_e)$ at high electron temperatures vary by orders of magnitude[32–34], and the sample thickness modulates the competition between electronic pressure relaxation and thermalization, our simulations show that these factors only quantitatively shift the melting thresholds and do not change the heterogeneous melting behaviors (Supplementary Figs. 8–10). Specifically, adopting a stronger electron-phonon coupling model ($G(T_e)$ from Lin et al. or Smirnov et al.) leads to a lower laser fluence threshold required to trigger ultrafast heterogeneous melting, as the lattice heats up more rapidly. While using a weaker coupling model (e.g., $G(T_e)$ from Medvedev et al.) yields melting dynamics and thresholds similar to those obtained with a constant electron-phonon coupling.

Similarly, reducing the sample thickness shortens the propagation time of electronic stress waves and still results in the emergence of pressure-driven heterogeneous melting fronts even in films as thin as

10 nm. For intermediate thicknesses such as 20 nm, our simulations show that the propagation time of stress waves becomes comparable to the thermalization timescale, leading to a competition between ultrafast heterogeneous melting and homogeneous melting. These results highlight that the emergence of electronic pressure from the laser-excited PES fundamentally modifies the melting dynamics, and this effect manifests similarly across a wide range of model parameters and sample geometry.

### Ubiquity of heterogeneous structural transformations

To demonstrate the ubiquity of electronic pressure in laser-driven processes, we extend the TTM-DPMD framework to gold (Au), a noble metal with fully occupied $d$-orbitals. Despite its distinct electronic structure, Au exhibits analogous nonthermal dynamics (Supplementary Fig. 11). At elevated electron temperatures, the hot electrons also contribute non-negligible electronic pressure in Au, reaching 25 GPa at $T_e = 20,000$ K. Under isobaric-isothermal conditions ($T_i = 300$ K, $p_z = 1$ bar), the laser-excited Au undergoes significant uniaxial distortion. As $T_e$ reaches 17,000 K, the expansion can even destabilize the lattice, triggering a disordered transition at room temperature ($T_i = 300$ K)—a "nonthermal melting" regime where amorphization occurs without lattice heating. These findings align with the conclusions of Daraszewicz et al. and Medvedev et al. regarding nonthermal behaviors under isobaric constraints[13,14], suggesting that the ultrafast disordering of laser-excited Au is primarily due to the electron pressure relaxation process.

Further, large-scale simulations of 35-nm-thick free-standing Au nanofilm confirm ultrafast heterogeneous melting (Supplementary Fig. 12), mirroring W's behavior. With the release of electronic pressure, the surface atoms directly transform into a disordered structure due to a significant reduction in melting point. The melt front then moves inward at a speed of 2916 m s$^{-1}$, completing the melting process within 8 ps. These results, along with previous predictions on the widespread existence of electronic pressure in laser-excited materials[11], underscore the pervasive nature of electron pressure relaxation dynamics and ultrafast heterogeneous structural transformation in laser-driven processes.

### Summary

By investigating the real-time response of laser-excited W and Au nanofilms via the TTM-DPMD approach, we unraveled the vital role of the electronic pressure in the laser-driven dynamics, which was previously not fully understood. As an integral part of nonthermal behavior, the electronic pressure builds up simultaneously with the appearance of the two-temperature state. At elevated electron temperatures, the additional pressure contributed by hot electrons becomes significant, reaching up to $p_e \sim 10^1$ GPa at $T_e \sim 10^4$ K. The release of electronic pressure triggers violent uniaxial expansion from the free surface, which exhibits a drastic difference from thermal expansion in temporal behaviors, which can be identified by time-resolved XRD techniques. As a result, pressure, density, and temperature discontinuity in the thermodynamic profile are generated and move inward at a high speed on the order of $10^3$ m s$^{-1}$.

The following structural transformation is therefore dominated by the relaxation of electronic pressure. Two unexpected phenomena, absent in either purely thermal process or previous lattice dynamics calculations, were observed: (i) heterogeneous solid-solid transition (in W), which directly arises from the expansion process that drives the BCC-FCC transition along the uniaxial Bain path. (ii) ultrafast heterogeneous melting (both in W and Au), which is a combination of density decrease driven by nonthermal expansion and temperature increase induced by electron-phonon coupling. These findings suggest an extreme structural heterogeneity in laser-excited metal, which has a profound influence on the interpretation of experiments and our comprehension of nonthermal behavior.

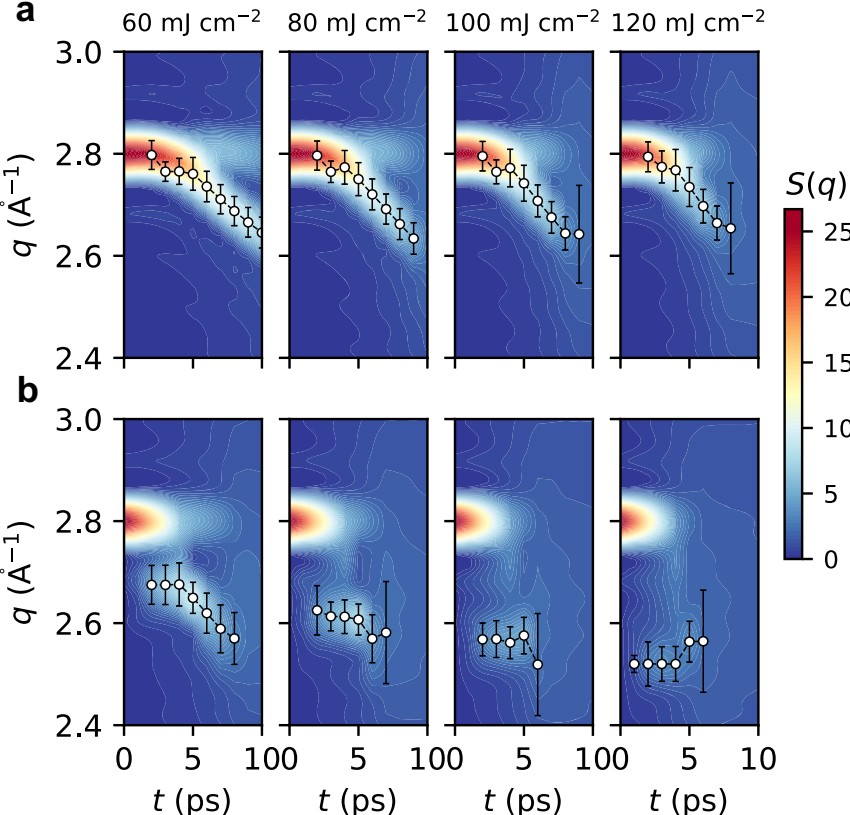

**Fig. 5 | Laser fluence dependence of thermal and nonthermal expansion dynamics.** Temporal evolution of static structure factor $S(q)$ of laser-excited W nanofilm at different laser fluence, estimated from TTM-DPMD trajectories with **a** ground-state PES and **b** laser-excited PES. The positions of the new diffraction peak, estimated by Gaussian fitting, are marked by black circles to highlight the difference between thermal and nonthermal expansion dynamics. The error bars represent the standard deviation of the Gaussian peak width.

We also note that this structural transformation is fundamentally different from the conventional nonthermal phase transition driven by lattice instability. The latter is expected to occur in less than a phonon period[8,35,36], which can be considered as homogeneous and ultrafast. Instead, the nonthermal expansion induces heterogeneous structural transformation, which can strongly rely on the crystal symmetry, the orientation of the free surface, and the strength of nonthermal pressure. Especially, even when phonon spectra suggest a dynamically stable structure, this structural transformation can occur. This is particularly evident in laser-excited Au.

Moreover, considering the interplay between this heterogeneous phase transition and ultrafast melting, whether the intermediate solid phase can be observed by experimental measurements depends on the accurate determination of electron-phonon coupling. Especially for the extremely inhomogeneous thermodynamic profile, both electron temperature and density dependence of the electron-phonon energy exchange rate should be considered in future work. On this issue, the TTM-DPMD approach can be coupled with the newly developed deep learning scheme to infer electronic structure on-the-fly[37,38].

## Methods
### Electron-temperature-dependent neural network model
In laser-excited metals, electron-electron scattering occurs on the femtosecond timescale[39], leading to ultrafast thermalization of excited electrons within $10^1$–$10^2$ fs. This results in a thermal equilibrium state of the electron subsystem with a well-defined temperature $T_e$. Since we focus on the ion dynamics for the picosecond timescale of interest, the nonthermal nature of laser-excited electrons in the early stages after

excitation is not considered here. In this context, "nonthermal" primarily denotes the non-equilibrium state where the electrons have a well-defined temperature $T_e$ that is different from the lattice temperature $T_i$. Therefore, the complex PES of laser-excited matter can be described within the framework of finite-temperature density functional theory[40], which is a free energy surface $A(\mathbf{R}, T_e)$ of an electron-ion system.

To capture the modification on PES introduced by hot electrons, here we adopt the deep neural network with an additional parameter, electron temperature $T_e$, named as electron-temperature-dependent deep potential model[12],

$$A = A(\mathbf{R}, T_e) = \sum_i \mathbf{N}_{\alpha_i}(\mathbf{D}_{\alpha_i}(r_i, \{r_j\}_{j \in n(i)}), T_e) \quad (1)$$

where $A(\mathbf{R}, T_e)$ indicates that the free energy depends on the local atomic environment $\mathbf{R}$ and electron temperature, $\mathbf{N}_{\alpha_i}$ denotes the neural network of specified chemical species of $\alpha_i$ of atom $i$, and the descriptor $\mathbf{D}_{\alpha_i}$ describes the local environment of atom $i$ with its neighbor list $n(i) = \{j | r_{ji} < r_{cut}\}$. The ETD-NN model can inherently incorporate the nonthermal characteristics of laser-excited matters in molecular dynamics simulations, including the hot-electron-modulated atomic forces and virial tensor,

$$\mathbf{F} = -\frac{\partial A}{\partial \mathbf{R}_i} = -\frac{\partial U}{\partial \mathbf{R}_i} + T_e \frac{\partial S}{\partial \mathbf{R}_i} \quad (2)$$

$$\boldsymbol{\Xi} = \frac{\partial A}{\partial \boldsymbol{\Omega}} \cdot \boldsymbol{\Omega}^T = \frac{\partial U}{\partial \boldsymbol{\Omega}} \cdot \boldsymbol{\Omega}^T - T_e \frac{\partial S}{\partial \boldsymbol{\Omega}} \cdot \boldsymbol{\Omega}^T \quad (3)$$

here $\mathbf{R}_i$ denotes the coordinate vector of atom $i$, $\boldsymbol{\Omega}$ denotes a $3 \times 3$ matrix formed by three lattice vectors, and $\boldsymbol{\Omega}^T$ is its transpose. Consider a system composed of $N$ atoms, the total pressure of this electron-ion system is determined by the summation of ionic thermal kinetic contribution and virial contribution.

In the main text, the ETD-NN model of tungsten is generated with DeePMD-kit packages (v2.2.5)[41] and DP-Generator (v0.10.0)[42]. The configuration space with different electronic occupations considered in this work is efficiently sampled. Based on the training data set generated in previous work[12], specifically, BCC structure (54 atoms) and liquid structure (54 atoms) are considered as the initial configurations and run DPMD under NVT and NPT ensemble (both isotropic and uniaxial constrains are considered), where temperatures ranges from 100 K to 6000 K, pressure ranges from −15 to 60 GPa, and corresponding electronic temperature ranges from 100 K to 25,000 K. The training sets consist of 6366 configurations under equilibrium conditions ($T_e = T_i$) and 6820 configurations sampled under two-temperature state ($T_e > T_i$). Moreover, we further improved the uniaxial expansion behavior of laser-excited tungsten along (100), (110) and (111) directions by sampling an additional 1860 configurations, where the ion temperature ranges from 100 K to 4000 K, electron temperature ranges from 100 K to 25,000 K.

For DP training, the embedding network is composed of three layers (25, 50, and 100 nodes) while the fitting network has three hidden layers with 240 nodes in each layer. The total number of training steps is set to 400,000. The radius cutoff $r_c$ is chosen to be 6.0 Å. The weight parameters in the loss function for energies $p_e$, forces $p_f$, and virials $p_V$ are set to (0.02, 1000, 0.02) at the beginning of training and gradually change to (1.0, 1.0, 1.0).

The self-consistency calculations are all performed with the VASP packages (v6.2.0)[43]. The Perdew-Burke-Erzerhof exchange correlation functional is used[44], and pseudopotential takes the projector augmented-wave formalism[45]. The sampling of the Brillouin zone is chosen as 0.2 Å$^{-1}$ under ambient conditions and 0.5 Å$^{-1}$ for high temperature ($T \geq 1600$ K).

## Two-temperature model coupled neural network molecular dynamics approach

To simulate the real-time response of material upon laser excitation, an additional description of the electron subsystem is introduced and strongly coupled with the atomic system in the TTM-DPMD approach. The heat conduction equation of electron continuum characterizes the temporal evolution of electron occupation, thus governing the transition of the atomic system between different $T_e$-dependent PES. Langevin dynamics is incorporated to mimic the dynamic electron-ion collision, thus including the non-adiabatic energy exchange between the electron and the atomic subsystem,

$$C_e(T_e)\frac{\partial T_e}{\partial t} = \nabla \cdot (\kappa_e \nabla T_e) - g_{ei}(T_e)(T_e - T_i) + S(\mathbf{r}, t) \quad (4)$$

$$m_i \frac{d^2 \mathbf{r}_i}{dt^2} = -\nabla_i A(T_e) - \gamma_i \mathbf{v}_i + \tilde{\mathbf{F}}_i(t) \quad (5)$$

where $C_e$ the electron heat capacity, $\kappa_e$ the electronic thermal conductivity, $g_{ei}$ the electron-phonon coupling constant, $S(\mathbf{r}, t)$ the laser source, $\gamma_i$ the friction parameter, $\tilde{\mathbf{F}}_i$ the random force with Gaussian distribution.

To simulate the material response relevant to real experimental conditions, here the 30-nm-thick W nanofilm[27] and 35-nm-thick Au nanofilm[18] are chosen as the target sample. A Gaussian temporal profile of a laser pulse is used, and the duration is set to 130 fs. Since both the thickness of W and Au nanofilms is comparable to the mean free path of excited electrons (~33 nm in W and ~100 nm in Au), the ballistic motion of the excited electrons leads to the fast (within ~100 fs)

redistribution of the deposited energy within the ballistic range. As a result, the uniform laser energy deposition is expected,

$$S(t) = \frac{F_{abs}}{\sigma d \sqrt{2\pi}} \exp\left(-\frac{(t-t_0)^2}{2\sigma^2}\right) \quad (6)$$

where $F_{abs}$ the absorbed laser fluence, $\sigma$ the standard deviation of the temporal Gaussian distribution, $d$ the thickness of nanofilm, $t_0$ the time zero defined as the arrival of the maximum of the laser pulse. The absorbed laser energy density $\epsilon$ can be estimated via $\epsilon = F_{abs}/\rho d$, where $\rho$ the mass density.

In the main text, the TTM-DPMD simulations are performed with LAMMPS packages (version 29 Sep 2021)[46] through modified EXTRA-FIX module. The electronic heat capacity is obtained by individual DFT calculations $C_e = T_e \frac{\partial S_e}{\partial T_e}$. The electron thermal conductivity is described by the Drude model relationship, $\kappa_e(T_e, T_i) = \frac{1}{3} v_F^2 C_e(T_e) \tau_e(T_e, T_i)$, where $v_F$ is Fermi velocity and $\tau_e(T_e, T_i)$ is the total electron scattering time defined by the electron-electron and electron-phonon scattering rates, $1/\tau_e = 1/\tau_{e-e} + 1/\tau_{e-ph} = AT_e^2 + BT_i$. The coefficients $A = 2.11 \times 10^{-4}$ K$^{-2}$ ps$^{-1}$, $B = 8.4 \times 10^{-2}$ K$^{-1}$ ps$^{-1}$, $v_F = 9710$ Å ps$^{-1}$ are adopted. Considering the accurate determination of electron-phonon coupling at elevated electron temperatures remains controversial, a constant electron-phonon coupling value ($G_0 = 2.0 \times 10^{17}$ W m$^{-3}$ K$^{-1}$) from Mo et al.'s experiments[27] is used in the main text. Furthermore, a detailed discussion with the different electron-temperature-dependent $G(T_e)$ predictions[32–34] is also provided; we refer to the "Discussion" section and Supplementary Materials (Supplementary Figs. 8 and 9).

For the atomic system, the sample geometry of 30-nm-thick 100-oriented single-crystalline W foil is a parallelepiped $L_x \times L_y \times L_z$ with $L_x = L_y = 30a_0$ and $L_z = 95a_0$ (171,000 atoms). $a_0 = 3.17104$ Å is the parameter of the elementary BCC cell for W corresponding to the ETD-NN model prediction at 300 K and 0 GPa. Periodic boundary conditions are applied in the $x$ and $y$ directions, while a free boundary condition is applied to the $z$-axis to allow the expansion of the free surface. The lateral periodic conditions ($x$ and $y$-axis) simulate the experimental situation in which the laser spot diameter is large (hundreds of microns) compared to the depth of the laser energy deposition (tens of nanometers), so that the effects of the edges of the laser beam can be neglected.

## Data availability

The relevant data generated in this study have been deposited in the Zenodo[47]. Source data are provided with this paper.

## Code availability

LAMMPS and DeePMD-kit are free and open source codes available at https://github.com/lammps/lammps and https://github.com/deepmodeling/deepmd-kit, respectively. VASP is a commercial code available from https://www.vasp.at. Detailed instructions for obtaining and using these codes can be found on their respective websites.

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

## Acknowledgements

This work was supported by the National Natural Science Foundation of China under Grant Nos. 12504326, 12534013, 12304307, 12122103, 12104507, the National University of Defense Technology Research Fund Project, the Science and Technology Innovation Program of Hunan Province under Grant No. 2025YJ001 and 2021RC4026.

## Author contributions

Q.Z. and J.D. proposed the original idea and designed the research. Q.Z. carried out the simulations, Q.Z., X.Y., D.K., and J.D. analyzed and interpreted the results. Q.Z., X.Y., and J.D. wrote the manuscript with the help of other authors. B.C., S.Z., and K.C. provided additional support for the interpretation of the results.

## Competing interests

The authors declare no competing interests.
