## [Transparent Peer Review file · Nature Communications]

Picosecond-scale Heterogeneous Melting of Metals At Extreme Non-equilibrium States

Corresponding Author: Professor Jiayu Dai

Version 0:

Reviewer comments:

Reviewer #1

(Remarks to the Author)

The manuscript by Zeng et al presents a very interesting study on how metal nanofoil melts under ultrafast laser heating. The group has used their recently-developed hybrid atomistic-continuum modeling approach, which is a two-temperature model coupled with deep potential molecular dynamics (TTM-DPMD) method, to simulate femtosecond laser heating and melting of W and Au nanofoils. What they found are several new phenomena: (a) ultrafast laser heating will drive thermal electrons to build up electronic pressure of tens of GPa inside the nanofoil; (b) The release of such high electronic pressure at the free surface can induce the foil expansion within ~ps timescale; (c) The ultrafast expansion will cause density dropping thereby leading to solid-solid phase transition (e.g., BCC to FCC for W); and (d) It can also drive a rapidly-moved melting front in the bulk material. The success of their work was based on the incorporation of electron temperature dependence for training potential energy surface (PES) with neuro-network in DPMD. The authors even provided the calculated XRD features for experiments to test their predictions. Overall, I find these results are scientifically sounded and their findings are important to the broad physics community who cares about how ultrafast laser-matter interactions. Most importantly, these results are significant to the broad audience of Nature Communications.

For these reasons, I do recommend a revised version of this manuscript for publication in Nature Communications. Here are my suggestions for the authors to consider if they choose to resubmit their revised manuscript to Nature Communications:

1. The authors used a constant value of 2×10^{17} W/m²K for the electron-phonon coupling, which was debated over the past decade for warm-dense matters. Some experiments/theories indicated this value should be on this high level, while others suggested several times lower than this value. I wonder how their results may vary with this choice; After all, electron-phonon coupling is the main path for energy exchange between the subsystems. They should give some justification about their choice; at least some discussions are guaranteed.
2. The authors need to clearly state what boundary condition they used for the z-axis direction (open or periodic?), even though they specified the periodic boundary condition for x/y-dimensions.
3. I feel that the Au results do not add much to the main context. I would suggest them to consider moving Figs. 6 and 7 of AU results to Supplementary Materials. However, this is just optional for the authors to consider.
4. Some typos and grammars need to be cleaned: (a) "TTD-DPMD" should be "TTM-DPMD" in the captions of Figs. 4 & 5; (b) "hot-electron-contributed" would better be "hot-electron-induced"; (c) The labels for Fig.1a might better be "TTM-MD (ground-state PES)" and Fig.1b's be "ETD-DPMD (laser-excited PES)"; (d) I gather that changing the word "modulation" to "modification" should better match what the authors meant; and (e) Some explanation is needed for the x-axis of Fig. 2(c); what is it?

Reviewed by Suxing Hu @ University of Rochester

Reviewer #2

(Remarks to the Author)

I reviewed the paper by Zeng et al. on the Ultrafast Heterogeneous Melting of Metals under ultrafast laser excitations. The authors used a combination of neural-network enhanced multi scale simulations, TTM-DPMD approach, molecular dynamics (LAMMPS) and ab initio methods (VASP). The main results consisted in the elucidation of pressure relaxation as one of the ingredients for heterogeneous phase transformations. The findings are interesting, and may generate several implications in the field of laser-driven phase transitions and nearby areas. However, the paper would need some additional work before being accepted in this journal. My comments are included below:

1) On figure 1: it is not clear what the author meant by 'homogeneous melting and heterogeneous'. They are at different parts of the local view of the structure but it is not clear their meaning. Moreover, if the authors could highlight in the evolution of the structure from 0 ps until 7 ps how much in Å's the systems expanded would be helpful.

The local view of the structure at some of the time steps would be important to be included in an additional row underneath panels 1a and 1b. For instance, from full BCC, until it reached FCC, and maybe at the transition point BCC-FCC. HCP is not clearly seen in the plot. The authors may consider to include it at somewhere to show the differences.

2) The caption in Figure 2 definitely needs additional details to be fully understandable. There are also several typos which would need revisions.

3) Is there any reason to include Figure 7 in the main text? It could easily be moved to SI, with a few comments included in the text.

4) Are expected any variations of the ultrafast heterogeneous process with film thickness? I can see that W and Au was considered at 33 nm and 100 nm, respectively, but I have the impression that these results might change if the thickness become too atomic (<10 nm). I'd suggest the authors include some discussions and possibly data demonstrating thickness effects, if any, in their paper. This would certainly increase the impact of the manuscript across different fields.

Reviewer #3

(Remarks to the Author)

This manuscript employs a computational model recently developed by the authors to describe laser-heated matter. I cannot support publication of this manuscript in ncomms for reasons I will detail below. The topic of the paper is ultrafast heterogeneous melting.

The manuscript is written in an overly vague way. There are "melting paradigms", "complex interplays", "intricacy of laser-driven process", and so on. It would be better to know precisely what these are referring to. It reads like it was written by AI, which tends to hype everything.

The authors argue that these systems are far from equilibrium, but the conditions they state are those of a 2T equilibrium. This is expected for all systems with an external energy source when there are disparate masses. This quasi-equilibrium occurs in shocked system, laser-driven systems (as is described here), systems heated with ion beams and so on. This state of matter is in fact quite common and has been studied for many decades. Of course, the authors know this; what isn't clear is why they use such hyperbole here. Hyperbole can only make the science worse by tempting readers to stop reading.

If the system is far from equilibrium one needs a kinetic model, not a low-order hydrodynamics model. Either the system is not far from equilibrium or the model is inapplicable. The authors state that the MFP of the electrons is of order the sample size (in the first subsection of Results) - with a $Kn > 1$, the models used here - in fact, the hydro notions of pressure - are not valid. I don't see how a temperature-dependent potential can describe non-thermal characteristics.

All of this is important because one needs to establish what the big discovery is that warrants a paper and one in this journal. This appears to be the "paradigm shift" due to hot electrons exerting an additional pressure and the question of how this affects structural transformations. From what I can tell, this means using temperature-dependent input, a rather obvious idea. (In fact, it isn't clear why this would not be done for any system that has an electron temperature near or greater than the Fermi energy.) What I was expecting to see was a "smoking gun" related to the "vital role of the electronic pressure". Since electronic pressure is always present, what we are looking for is some comparison of the electron pressure relative to other terms spatially. The fact that there are more terms in a heterogeneous system is trivial - of course, spatial gradient terms become important when there are spatial gradients. That the electronic pressure is "vital" requires more effort to show. I always think of electronic pressure as being important: it drives many phenomena across a wide range of materials; for example, it is responsible for ambipolar diffusion in plasmas. The visualizations of the simulations can only partly explain this phenomena: back of the envelope, perhaps Thomas-Fermi based, to show the underlying physics in a simple way.

Because of the heterogeneity of the system, and the large electron MFP, we expect the free energy A to have time-evolving spatial gradients. How was the ETD-DP trained on various temperatures densities and each of their gradients? Or, was A trained on homogeneous systems and used in a "slowly varying" approximation here by using a local density in A ?

In summary, I do not recommend publication of this manuscript. Apart from the writing and unclear descriptions, it is not clear that a significant problem has been solved. The manuscript reads as series of toy experiments with simple observations based on extracted observables. The experiments don't entirely focus on the main theme and the observables don't prove

the point the authors appear to be interested in.

Version 1:

Reviewer comments:

Reviewer #2

(Remarks to the Author)

The authors have addressed all my comments correctly, and I am satisfied with the revisions. However, I'd only remark that Figure 2 would be better displayed if the panels are oriented vertically instead of horizontally. For instance, like in the scheme below.

A. B.
C. D
E. F

Moreover, the panels currently in 'C' would need to be labelled and be at similar size as panels A and B. At the moment, panels in C look very small, and we barely can see the details. After that, the paper can be accepted for publication.

Reviewer #3

(Remarks to the Author)

The authors have done a fine job of improving the manuscript and I support publication.

Response

(Dated: June 9, 2025)

Manuscript ID: NCOMMS-25-13275

Title: Ultrafast Heterogeneous Melting of Metals under Extreme Non-equilibrium States

Dear Editors and Reviewers,

We sincerely appreciate the constructive comments and valuable suggestions provided by you and all reviewers. In response, we have carefully revised the manuscript to address each point raised. The main improvements are summarized as follows:

1. The Discussion and Supplementary Information have been expanded to provide a more detailed analysis on the influence of electron-phonon coupling parameter $G(T_e)$ and sample thickness d on laser-induced melting, including additional simulations with different $G(T_e)$ models (Fig.S8 and Fig.S9) and different sample thicknesses (Fig.S10), as recommended by Reviewer #1 and Reviewer #2.

2. We have systematically revised the scientific terminology and descriptions throughout the manuscript to improve clarity and avoid ambiguity or overstatement, as suggested by Reviewer #3.

3. The Au results and related figures (Fig.6 and Fig.7) have been moved to the Supplementary Information (Fig.S11 and Fig.S12), and the discussion in the main text has been streamlined, as suggested by both Reviewer #1 and Reviewer #2.

4. Figures and captions (especially for Fig.1 and 2) have been revised for improving clarity and completeness, and typos have been corrected as pointed out by the reviewers.

5. We have clarified the boundary conditions used in the simulations and revised the relevant sections in the Methods, following Reviewer #1's suggestion.

We hope these substantial revisions have significantly improved the quality, clarity, and scientific rigor of our manuscript. We are grateful for the reviewers' insightful feedback, which has been invaluable in strengthening our work.

REPLY TO REVIEWER #1

Reviewer #1 (Remarks to the Author):

The manuscript by Zeng et al presents a very interesting study on how metal nanofoil melts under ultrafast laser heating. The group has used their recently-developed hybrid atomistic-continuum modeling approach, which is a two-temperature model coupled with deep potential molecular dynamics (TTM-DPMD) method, to simulate femtosecond laser heating and melting of W and Au nanofoils. What they found are several new phenomena: (a) ultrafast laser heating will drive thermal electrons to build up electronic pressure of tens of GPa inside the nanofoil; (b) The release of such high electronic pressure at the free surface can induce the foil expansion within ps timescale; (c) The ultrafast expansion will cause density dropping thereby leading to solid-solid phase transition (e.g., BCC to FCC for W); and (d) It can also drive a rapidly-moved melting front in the bulk material. The success of their work was based on the incorporation of electron temperature dependence for training potential energy surface (PES) with neuro-network in DPMD. The authors even provided the calculated XRD features for experiments to test their predictions. Overall, I find these results are scientifically sounded and their findings are important to the broad physics community who cares about how ultrafast laser-matter interactions. Most importantly, these results are significant to the broad audience of Nature Communications. For these reasons, I do recommend a revised version of this manuscript for publication in Nature Communications.

Reply:

We are grateful to the reviewer for the thorough reading of our manuscript and providing valuable feedback. We sincerely appreciate the constructive comments and recommendations. According to your advice, we greatly improved our manuscript and added a lot of useful information.

1. The authors used a constant value of $2 \times 10^{17} \text{W/m}^3 \text{K}$ for the electron-phonon coupling, which was debated over the past decade for warm-dense matters. Some experiments/theories indicated this value should be on this high level, while others suggested several times lower than this value. I wonder how their results may vary with this choice; After all, electron-phonon coupling is the main path for energy exchange between the subsystems. They should

give some justification about their choice; at least some discussions are guaranteed.

Reply:

Thank you for your valuable advice. As you rightly pointed out, the electron-phonon coupling serves as a critical role in non-equilibrium dynamics but remains on highly debated, where existing studies predict the order-of-magnitude discrepancies at elevated electron temperatures ($T_e \geq 10^4$ K).

Following your advice, we have conducted further investigations into how different electron-phonon coupling models $G(T_e)$ affect the key physical process (laser induced melting dynamics) discussed in the manuscript, which supplements our understanding of this topic. These findings are comprehensively discussed in the *Discussion* Section of the main text and supported by extended analysis in Supplementary Materials *Section E* (Fig.S8 and Fig.S9).

The brief summary is presented on page 6 of the revised manuscript, as follows:

”We further examined the effects of electron-phonon coupling strength $G(T_e)$ and sample thickness on the ultrafast heterogeneous melting behavior. Although theoretical predictions for $G(T_e)$ at high electron temperatures vary by orders of magnitude, ..., our simulations show that these factors only quantitatively shift the melting thresholds and does not change the heterogeneous melting behaviors (Fig.S9 and Fig.S10). Specifically, **adopting a stronger electron-phonon coupling model ($G(T_e)$ from Lin *et al.* or Smirnov *et al.*) leads to a lower laser fluence threshold required to trigger ultrafast heterogeneous melting, as the lattice heats up more rapidly. While using a weaker coupling model (e.g., $G(T_e)$ from Medvedev *et al.*) yields melting dynamics and thresholds similar to those obtained with a constant electron-phonon coupling ...** These results highlight that the emergence of electronic pressure from the laser-excited potential energy surface fundamentally modifies the melting dynamics, and this effect manifests similarly across a wide range of model parameters and sample geometry. ”

A detailed discussion is on page 5-8 of the revised Supplementary Materials, as follows:

”Fig.S8(a) summarizes experimental and theoretical $G(T_e)$ values. Experimental measurements at room temperature yield $G_0 = (2.0 \pm 0.5) \times 10^{17}$ Wm⁻³K⁻¹ (UED experimental measurement by Mo *et al.*) and $G_0 = (1.4 \pm 0.3) \times 10^{17}$ Wm⁻³K⁻¹ (optical pump-probe measurements by Daraszewicz *et al.*). Theoretical predictions diverge significantly at high electron temperatures: at $T_e = 20,000$ K, Lin *et al.* and Smirnov *et al.* predict

$\sim 12.0 \times 10^{17} \text{ Wm}^{-3}\text{K}^{-1}$ and $\sim 7.0 \times 10^{17} \text{ Wm}^{-3}\text{K}^{-1}$, respectively, while Medvedev *et al.* suggest a relatively lower value $\sim 2.0 \times 10^{17} \text{ Wm}^{-3}\text{K}^{-1}$.

FIG. S8: Influence of $G(T_e)$ on electron-phonon energy relaxation. (a) Electron-temperature dependence of electron-phonon coupling strength, where circles/squares denote experimental measurements, and solid lines represent theoretical predictions. (b) Temporal evolution of electron temperature T_e (dashed lines) and ion temperature T_i (solid lines) obtained by TTM-DPMD simulations by adopting different $G(T_e)$. Black dotted line indicates the isochoric melting point of laser-excited tungsten.

Influence of $G(T_e)$ on electron-phonon energy relaxation. Fig.S8(b) shows temperature evolution from TTM-DPMD simulations. Stronger coupling models ($G(T_e)$ from Lin *et al.* or Smirnov *et al.*) induce rapid lattice heating, surpassing the isochoric melting point $T_{m,\rho_0}(T_e)$ within 3-5 ps. In contrast, a weaker coupling model ($G(T_e)$ taken from Medvedev *et al.*) exhibits heating rates comparable to constant electron-phonon coupling approximations (G_0 from Mo *et al.*'s measurement) in the main text. This results in slower thermalization where the lattice temperature exceeds the isochoric melting point within 15 ps at $F_{abs} = 60 \text{ mJ/cm}^2$ and 7 ps at $F_{abs} = 120 \text{ mJ/cm}^2$ respectively.

Influence of $G(T_e)$ on melting dynamics. Fig.S9 illustrates the temporal evolution of atomic configurations. In Fig.S9(a)(b), when using stronger $G(T_e)$ model from Lin *et al.*'s work, moderate fluence (60 mJ cm^{-2}) provides a comparable timescales of heating and pressure relaxation, enabling ultrafast heterogeneous melting. For laser fluence above 100 mJ cm^{-2} , accelerated lattice heating induces homogeneous nucleation of liquid phase in bulk regions, completing melting prior to stress wave propagation, indicating a homogeneous melting mechanism. For $G(T_e)$ from Smirnov *et al.*, the behavior is qualitatively similar to Lin *et al.*'s $G(T_e)$ (Fig.S9(c)(d)).

FIG. S9: Influence of $G(T_e)$ on melting dynamics. (a-f) temporal evolution of atomic configurations. Red dashed lines mark the melt fronts. Liquid local structure (gray atoms) were determined using PTM method.

Medvedev *et al.*'s weaker $G(T_e)$ reveals similar behavior to the constant G_0 case presented in the main text, see Fig.S9(e)(f). Melting initiates and completes below the isochoric melting threshold $T_i < T_{m,\rho_0}(T_e)$ across all laser fluences, indicating a dominant role of electron pressure relaxation in driving melting process. At higher fluences ($> 100 \text{ mJ cm}^{-2}$), surface density decrease combined with rapid heating drives surface-initiated ultrafast heterogeneous melting.”

2.The authors need to clearly state what boundary condition they used for the z-axis direction (open or periodic?), even though they specified the periodic boundary condition for x/y-dimensions.

Reply:

Thank you for pointing this out. We apologize for the misleading, we revised the last paragraph in the *Method* section, as follows:

”For atomic system, the sample geometry of 30-nm-thick 100-oriented single-crystalline W foil is a parallelepiped $L_x \times L_y \times L_z$ with $L_x = L_y = 30a_0$ and $L_z = 95a_0$ (171,000 atoms). $a_0 = 3.17104 \text{ \AA}$ is the parameter of the elementary BCC cell for W corresponding to the ETD-NN model prediction at 300 K and 0 GPa. Periodic boundary conditions are applied in the x and y directions, **while free boundary condition is applied to z-axis to allow the expansion of free surface**. The lateral periodic conditions (x and y -axis) simulate the experimental situation in which the laser spot diameter is large (hundreds of microns) compared to the depth of the laser energy deposition (tens of nanometers) so that the effects of the edges of the laser beam can be neglected. ”

3.I feel that the Au results do not add much to the main context. I would suggest them to consider moving Figs. 6 and 7 of AU results to Supplementary Materials. However, this is just optional for the authors to consider.

Reply:

Thank you for your advice. In the *Discussion* section, we included the Au results to show that electron pressure relaxation dynamics work similarly in different metals. To improve clarity, we simplified the discussion about Au results and moved detailed analyses (Fig. 6-7) to Supplementary Materials, see *Section F Nonthermal response of laser-excited gold*.

The simplified discussion (on Page 6 of the revised manuscript) is as follows:

”**Ubiquity of heterogeneous structural transformations.** To demonstrate the ubiquity of electronic pressure in laser-driven processes, we extend the TTM-DPMD framework to gold (Au), a noble metal with fully occupied d -orbitals. Despite its distinct electronic structure, Au exhibits analogous nonthermal dynamics. At elevated electron temperatures, the hot electrons also contribute non-negligible electronic pressure in Au, reaching 25 GPa at $T_e = 20,000$ K (Fig.S11). Under isobaric-isothermal condition ($T_i = 300$ K, $p_z = 1$ bar), the laser-excited Au undergoes significant uniaxial distortion. As T_e reaches 17,000 K, the expansion can even destabilize the lattice, triggering a disordered transition at room temperature ($T_i = 300$ K)—a “nonthermal melting” regime where amorphization occurs without lattice heating. These findings align with the conclusions of Daraszewicz *et al.*, and Medvedev *et al.* regarding nonthermal behaviors under isobaric constraints, suggesting that the ultrafast disordering of laser-excited Au is primarily due to electron pressure relaxation process.

Further, large-scale simulations of 35-nm-thick free-standing Au nanofilm confirm ultrafast heterogeneous melting (see Fig.S12), mirroring W’s behavior. With the release of electronic pressure, the surface atoms directly transforms into a disordered structure due to significant reduction in melting point. The melt front then moves inward at a speed of $2,916 \text{ m s}^{-1}$, completing the melting process within 8 ps. These results, along with previous prediction on the widespread existence of electronic pressure in laser-excited materials, underscore the pervasive nature of electron pressure relaxation dynamics and ultrafast heterogeneous structural transformation in laser-driven processes. ”

4. Some typos and grammars need to be cleaned: (a) “TTD-DPMD” should be “TTM-DPMD” in the captions of Figs. 4 & 5; (b) “hot-electron-contributed” would better be “hot-electron-induced”; (c) The labels for Fig.1a might better be “TTM-MD (ground-state PES)” and Fig.1b’s be “ETD-DPMD (laser-excited PES)”; (d) I gather that changing the word “modulation” to “modification” should better match what the authors meant; and (e) Some explanation is needed for the x-axis of Fig. 2(c); what is it?

Reply:

Thank you for your suggestions. We have carefully revised the manuscript to address all the points you raised:

(a, b, d) Minor issues such as typos, terminology, and word choices (e.g., ”TTD-DPMD” to ”TTM-DPMD”, ”hot-electron-contributed” to ”hot-electron-induced”, and ”modulation”

to "modification") have all been corrected throughout the manuscript.

(c) The labels for Fig. 1a and 1b have been updated to "TTM-MD (ground-state PES)" and "TTM-DPMD (laser-excited PES)", respectively. We chose to use "TTM-DPMD" instead of "ETD-DPMD" to maintain consistency with the terminology used throughout the manuscript, as "TTM-DPMD" refers to our hybrid simulation framework that incorporates the electron-temperature-dependent potential. We hope this clarification will help readers clearly distinguish between the conventional and laser-excited simulation approaches.

(e) We have clarified in the caption of Fig. 2(c) that the x-axis represents the radial distance r used for the calculation of the radial distribution function and coordination number.

We have also proofread the manuscript to correct other minor typos and improve clarity. Thank you again for your helpful comments.

REPLY TO REVIEWER #2

Reviewer #2 (Remarks to the Author):

I reviewed the paper by Zeng et al. on the Ultrafast Heterogeneous Melting of Metals under ultrafast laser excitations. The authors used a combination of neural-network enhanced multi scale simulations, TTM-DPMD approach, molecular dynamics (LAMMPS) and ab initio methods (VASP). The main results consisted in the elucidation of pressure relaxation as one of the ingredients for heterogeneous phase transformations. The findings are interesting, and may generate several implications in the field of laser-driven phase transitions and nearby areas. However, the paper would need some additional work before been accepted in this journal.

Reply:

We are grateful to the reviewer for the thorough reading of our manuscript and providing valuable feedback. We sincerely appreciate the constructive comments and recommendations. According to your advice, we greatly improved our manuscript and added a lot of useful information.

1. On figure 1: it is not clear what the author meant by 'homogeneous melting and heterogeneous'. They are at different parts of the local view of the structure but it is not clear their meaning. Moreover, if the authors could highlight in the evolution of the structure from 0 ps until 7 ps how much in Å's the systems expanded would be helpful. The local view of the structure at some of the time steps would be important to be included in an additional row underneath panels 1a and 1b. For instance, from full BCC, until it reached FCC, and maybe at the transition point BCC-FCC. HCP is not clearly seen in the plot. The authors may consider to include it at somewhere to show the differences.

Reply:

Thank you for your constructive suggestions. The 'homogeneous melting' refers to uniform nucleation and growth of liquid-phase within the solid matrix, while 'heterogeneous melting' involves a well-defined solid-liquid interface formation (typically at surfaces) and propagation.

According to your suggestion, we have revised the Fig.1 to better elucidate the physical processes, including : (1) directional arrows are added in the enlarged views to highlight spatial characteristics of homogeneous melting and heterogeneous melting. (2) temporal evolution of sample thickness are added to quantify the nonthermal expansion dynamics.

We also note that the temporal evolution of sample thickness can be extracted from thermodynamic profiles in Supplementary Fig.S2, and surface expansion velocities are also detailed in Supplementary Fig.S3.

Regarding the detailed analysis on microscopic structural transformation, we primarily present this in Fig.2, which has also been revised to include time-resolved radial distribution functions (RDF) and coordination number (CN) for selected regimes. Specific details are provided in the *response to the Question 2*. We hope these revisions can strengthen the mechanistic interpretation while maintaining quantitative rigor.

Moreover, regarding the HCP structure, we would like to clarify that the HCP-type atoms mainly arise from thermally activated defects or local distortions in the BCC region, as well as from partial atomic slips within FCC domains. However, these HCP-identified atoms are distributed sparsely and do not form any significant bulk regions. Therefore, the overall phase transition process is dominated by the BCC-FCC transformation and subsequent melting. We have added a specific discussion in the Supplementary Information relating to Fig.S4 to clarify this point and avoid possible misunderstandings.

”We also note that, due to the characteristics of the PTM method, a small fraction of atoms are identified as HCP during the structural evolution. These HCP-type atoms mainly arise from thermally activated defects or local distortions in the BCC region, as well as from partial atomic slips within FCC domains. However, these HCP-identified atoms are distributed sparsely and do not form any significant bulk regions. Consequently, they do not contribute to long-range structural measurements such as X-ray diffraction. Therefore, the overall phase transition process is dominated by the BCC-FCC transformation and subsequent melting.”

2. The caption in Figure 2 definitely needs additional details to be fully understandable. There are also several typos which would need revisions.

Reply:

Thank you for your valuable advice. We have revised Fig.2 and its caption to better

Fig. 1: **Diverging melting dynamics in laser-excited tungsten.** Under absorbed laser fluence of 120 mJ cm^{-2} , (a) Conventional TTM-MD prediction with ground-state PES $A(\mathcal{R})$ showing homogeneous melting governed by electron-phonon coupling. (b) Our TTM-DPMD results with laser-excited PES $A(\mathcal{R}, T_e)$ revealing electronic pressure-driven heterogeneous melting. The local structures are identified by the polyhedral template matching (PTM) method, and polyhedral surface meshes around FCC-type (green) and amorphous-type (gray) particles are constructed to highlight the heterogeneity in lattice symmetry. Red arrows indicate the propagation direction of phase transition (solid-liquid or BCC-FCC) interfaces.

illustrate the difference of thermodynamic states and microscopic structure between interior and surface regions after laser excitation. The key improvements include:

(1) The temporal evolution of temperature, pressure, and density are only presented for interior ($z = 14.2 \text{ nm}$) and surface regions ($z = 1.8 \text{ nm}$), with specific time markers highlighting critical states (see Fig.2a and Fig.2b). This enables clear view of thermodynamic evolution in (p, T) and (ρ, T) spaces. During the first 3 ps, we can see the surface region experiences dramatic density decrease due to electronic pressure relaxation, while the interior region maintains isochoric ($\rho = \rho_0$).

(2) The detailed local structural evolution in both interior and surface regions are presented in Fig.2c, by providing time-resolved radial distribution functions (RDF) and coordination number (CN). Combined with Fig.2b, these characterizations elucidate the complete

Fig. 2: Extreme heterogeneity in laser-excited W. (a,b) Depth-dependent thermodynamic pathways in tungsten nanofilm under laser fluence of 120 mJ cm^{-2} . Colored circles show thermodynamic states at 0.5 ps intervals, with arrows indicating evolution trajectories. Red stars mark three representative thermodynamic states of surface region ($z = 1.8 \text{ nm}$) corresponding to: initial electronic pressure buildup ($t = 0.5 \text{ ps}$), complete pressure release ($t = 1.5 \text{ ps}$), and surface melting ($t = 3 \text{ ps}$). Blue stars denote simultaneous thermodynamic states of interior region ($z = 14.2 \text{ nm}$) for comparison. The calculated equilibrium isochore (gray solid line) and melting curve (black solid line) are also presented for comparison. (c) Radial distribution functions $g(r)$ and coordination numbers (CN) for surface and interior regions at selected time delays, where radial distribution of CN are obtained by integration of $g(r)$. The colored region in CN marks the radius of first shell.

transformation sequence of surface region: initial nonthermal expansion induces BCC-FCC transition ($t = 1.5 \text{ ps}$), followed by melting process due to lattice heating ($t = 3.0 \text{ ps}$).

3. Is there any reason to include Figure 7 in the main text? It could easily be moved to SI, with a few comments included in the text.

Reply:

Thank you for your advice. In the *Discussion* section, we included the Au results to show that electron pressure relaxation dynamics work similarly in different metals. To improve clarity, we simplified the discussion about Au results and moved detailed analyses (Figs. 6-7) to Supplementary Materials.

The simplified discussion (on Page 6 of the revised manuscript) is as follows:

”**Ubiquity of heterogeneous structural dynamics.** To demonstrate the ubiquity of electronic pressure in laser-driven processes, we extend the TTM-DPMD framework to gold (Au), a noble metal with fully occupied d -orbitals. Despite its distinct electronic structure, Au exhibits analogous nonthermal dynamics. At elevated electron temperatures, the hot electrons also contribute non-negligible electronic pressure in Au, reaching 25 GPa at $T_e = 20,000$ K (Fig.S11). Under isobaric-isothermal condition ($T_i = 300$ K, $p_z = 1$ bar), the laser-excited Au undergoes significant uniaxial distortion. As T_e reaches 17,000 K, the expansion can even destabilize the lattice, triggering a disordered transition at room temperature ($T_i = 300$ K)—a “nonthermal melting” regime where amorphization occurs without lattice heating. These findings align with the conclusions of Daraszewicz *et al.*, and Medvedev *et al.* regarding nonthermal behaviors under isobaric constraints, suggesting that the ultrafast disordering of laser-excited Au is primarily due to electron pressure relaxation process.

Further, large-scale simulations of 35-nm-thick free-standing Au nanofilm confirm ultrafast heterogeneous melting (see Fig.S12), mirroring W’s behavior. With the release of electronic pressure, the surface atoms directly transforms into a disordered structure due to significant reduction in melting point. The melt front then moves inward at a speed of $2,916 \text{ m s}^{-1}$, completing the melting process within 8 ps. These results, along with previous prediction on the widespread existence of electronic pressure in laser-excited materials, underscore the pervasive nature of electron pressure relaxation dynamics and ultrafast heterogeneous structural transformation in laser-driven processes. ”

4. Are expected any variations of the ultrafast heterogeneous process with film thickness? I can see that W and Au was considered at 33 nm and 100 nm, respectively, but I have the impression that these results might change if the thickness become too atomic (<10 nm). I’d suggest the authors include some discussions and possibly data demonstrating thickness effects, if any, in their paper. This would certainly increase the impact of the manuscript across different fields.

Reply:

Thank you for your valuable advice. As you rightly pointed out, the sample thickness plays a crucial role in affecting the relative timescales of electronic pressure relaxation and lattice thermalization, therefore resulting in variation of structural transformation dynamics.

Following your advice, we have conducted further investigations into how sample thickness affect the ultrafast heterogeneous process discussed in our manuscript. Interestingly, the TTM-DPMD simulation results revealed several important findings that enhance our understanding on this topic. These findings are summarized in the *Discussion* Section of the main text and supported by extended analysis in Supplementary Materials *Section F* (Fig.S10).

Besides, we would like to clarify that in our study, the W and Au film thicknesses are 30 nm and 35 nm respectively, while the electron mean free paths in W/Au are 33 nm and 100 nm. This implies that despite the optical penetration depth of the laser, ballistic electron transport redistributes laser energy on femtosecond timescales, leading to spatially-uniform energy deposition along the incident direction (i.e., homogeneous electron temperature distribution).

A detailed discussion is on page 7-8 of the revised Supplementary Materials, as follows:

”The ultrafast heterogeneous melting process is intrinsically related to the propagation of electronic stress waves, making the film thickness d a critical factor in understanding structural transformation dynamics. Given tungsten’s electron mean free path of ~ 33 nm, we conducted supplementary simulations for 10 nm and 20 nm thick W films to investigate thickness effects under uniform energy deposition. All TTM-DPMD parameters remained unchanged as compared with the 30 nm case in the main text. For consistent analysis across different geometries, we normalized the laser fluence F_{abs} to thickness-independent laser energy density ϵ through $\epsilon = F_{abs}/\rho d$.

Since the propagation velocity v of electronic stress waves is thickness-independent, the propagation time $\tau = d/v$ reduces for thinner films. Taken laser energy density of 2.10 MJ/kg as example, for the 30 nm system, electronic stress waves require approximately 4.6 ps to propagate from surface $z = 0$ nm to interior $z = d/2 = 15$ nm, while the 10 nm system completes this process within 1.5 ps. In 10 nm films (Fig.S10(a)), this reduced timescale makes electron-phonon energy exchange insufficient to reach melting condition before the stress waves converge. When the counter-propagating stress waves cross each other at 2 ps, the accumulated uniaxial distortion leads to complete BCC-FCC transformation. Subsequent outward propagation of stress waves induces further density decrease, ultimately initiating heterogeneous melting that propagates outward at ~ 3000 m/s.

While in 20 nm system (Fig.S10(b)), the characteristic time for dual surface-initiated

FIG. S10: Influence of sample thickness on melting dynamics. (a-c) Spatio-temporal evolution of atomic configurations under laser energy density of 2.10 MJ/kg. Red dashed lines mark the melt fronts. Local liquid structure N_{liq} were determined using PTM method.

stress waves to converge at the geometric center ($\sim 3 \text{ ps}$) becomes comparable to the thermalization timescale (2.5 ps) required to exceed the isobaric melting point. As a result, the system exhibits features of both ultrafast heterogeneous melting and homogeneous melting mechanism.

While in the 30 nm system, sufficient lattice heating occurs prior to stress wave arrival at the center (Fig.S10(c)). Under laser energy density of 2.10 MJ/kg, thermal activation in surface low-density regions initiates melting before convergence of stress waves, creating surface-initiated ultrafast heterogeneous melting phenomenon.”

The brief summary is presented on page 6 of the revised manuscript, as follows:

”We further examined the effects of electron-phonon coupling strength $G(T_e)$ and sample thickness on the ultrafast heterogeneous melting behavior. Although... the sample thickness

modulates the competition between electronic pressure relaxation and thermalization, our multiscale simulations show that these factors only quantitatively shift the melting thresholds and does not change the heterogeneous melting behaviors (Fig.S8-S10)... Similarly, **reducing the sample thickness shortens the propagation time of electronic stress waves, and still results in the emergence of pressure-driven heterogeneous melting fronts even in films as thin as 10 nm. For intermediate thickness such as 20 nm, our simulations show that the propagation time of stress waves becomes comparable to the thermalization timescale, leading to a competition between ultrafast heterogeneous melting and homogeneous melting.** These results highlight that the emergence of electronic pressure from the laser-excited potential energy surface fundamentally modifies the melting dynamics, and this effect manifests similarly across a wide range of model parameters and sample geometry. ”

REPLY TO REVIEWER #3

Reviewer #3 (Remarks to the Author):

This manuscript employs a computational model recently developed by the authors to describe laser-heated matter. I cannot support publication of this manuscript in ncomms for reasons I will detail below. The topic of the paper is ultrafast heterogeneous melting.

Reply:

We are truly grateful for your comments and suggestions, which have greatly improved the quality and scientific rigor of our work. We are sorry for the confusion in the previous manuscript, and we hope that the present substantial revisions can address your concerns and enhance the clarity and impact of our manuscript. Below, we provide point-by-point responses to each of your specific comments.

1. The manuscript is written in an overly vague way. There are "melting paradigms", "complex interplays", "intricacy of laser-driven process", and so on. It would be better to know precisely what these are referring to. It reads like it was written by AI, which tends to hype everything.

Reply:

Thank you for your valuable feedback regarding to the clarity of our manuscript. We admitted that some expressions, such as "melting paradigms" and "complex interplays," may have appeared vague or general. In the revised manuscript, we have carefully rephrased these descriptions to be more specific and concrete.

For example, the first sentence in the *Abstract* related to "melting paradigms" is revised as follows:

"Extreme electron-ion non-equilibrium states, generated by ultrafast laser excitation, lead to melting processes that are fundamentally different from those under conventional thermal equilibrium and remain not fully understood."

The "paradigm shift" in 2nd paragraph of *Introduction* is revised to "a significant shift in understanding materials response at picoseconds timescale".

The 3rd paragraph in *Introduction* relating to "intricacy of laser-driven process" is revised as follows:

"Addressing this issue is challenging because the laser-driven process involves several

coupled physical processes across different scales, including laser excitation that modifies the PES, non-adiabatic electron-ion energy exchange, and atomic-scale structural responses.”

In conclusion, we have systematically revised the text to avoid ambiguous or exaggerated language and to ensure that all scientific terms are clearly defined. We appreciate your suggestion, which has greatly helped us improve the manuscript’s precision and readability.

2. The authors argue that these systems are far from equilibrium, but the conditions they state are those of a 2T equilibrium. This is expected for all systems with an external energy source when there are disparate masses. This quasi-equilibrium occurs in shocked system, laser-driven systems (as is described here), systems heated with ion beams and so on. This state of matter is in fact quite common and has been studied for many decades. Of course, the authors know this; what isn’t clear is why they use such hyperbole here. Hyperbole can only make the science worse by tempting readers to stop reading.

Reply:

Thank you for your valuable comments, we would like to discuss this interesting topic with you with more details. We agreed with you that the two-temperature (2T) state is a well-established quasi-equilibrium regime (for electron or ion subsystems respectively) in many driven systems. Our intention was not to overstate the novelty of this state, but rather to emphasize **the specific effects on structural transformations arising from hot electrons in 2T states**, which raised lots of attention recently but remains not fully understood.

Moreover, we would also like to clarify that, in the field of laser–metal interactions and related high-energy-density physics, it is universally recognized to refer to the 2T state as a “nonequilibrium” state. This terminology is widely used in the recent literature, and our usage follows this established convention. For example, R. Ernstorfer et al., [Science 323, 1033-1037 (2009)] state in their introduction: *”As visible and near-infrared laser light is absorbed by the electrons, intense short-pulsed optical excitation **initially forms highly nonequilibrium conditions (that is, very hot electrons within a cool lattice)**...”* Similarly, Plettenberg et al. [Commun. Mater. 4, 63 (2023).], refer to *”A femtosecond laser pulse primarily interacts with the electrons and leaves the ions almost unaffected, leading to **an extreme non-equilibrium state of hot electrons at an increased temperature T_e and cold ions**...”*. Other works, such as in the references of Giret et al., [Phys. Rev. B

90, 094103 (2014)], and B. Mahieu et al., [Nat Commun 9, 3276 (2018)], also use “nonequilibrium” or “out-of-equilibrium” to describe the 2T states and the related dynamics.

We recognize that in a broad range of fields, “nonequilibrium” may be reserved for situations different from 2T state, and this difference in terminology may have led to confusion. We have revised any language that could be interpreted as hyperbolic. Moreover, we have clarified our focus on **non-equilibrium between electrons and ions, .ie., “the hot electrons and cold lattices”** instead of **nonequilibrium electrons**, and the resulting modifications to the potential energy surface and phase transformation pathways, as in the *Method* section in the revised manuscript,

”In laser-excited metals, electron-electron scattering occurs on the femtosecond timescale, leading to ultrafast thermalization of excited electrons within a few tens fs. This results in a thermal equilibrium state of electron subsystem with a well-defined temperature T_e . Since we focus on the ion dynamics for ps timescale of interest, the nonthermal nature of laser-excited electrons in the early stages after excitation is not considered here. In this context, ‘nonthermal’ primarily denotes the non-equilibrium state where the electrons have a well-defined temperature T_e that is different from the lattice temperature T_i . Therefore, the complex PES of laser-excited matter can be described within the framework of finite-temperature density functional theory, which is a free energy surface $A(\mathcal{R}, T_e)$ of an electron-ion system.”

We thank the reviewer for highlighting this point and helping us improve the scientific rigor and clarity of our presentation.

3.If the system is far from equilibrium one needs a kinetic model, not a low-order hydrodynamics model. Either the system is not far from equilibrium or the model is inapplicable. The authors state that the MFP of the electrons is of order the sample size (in the first subsection of Results) - with a $Kn > 1$, the models used here - in fact, the hydro notions of pressure - are not valid. I don't see how a temperature-dependent potential can describe non-thermal characteristics.

Reply:

Thank you very much for your insightful comment. As we have clarified in our response to the question above, we fully agreed with the reviewer that the “nonequilibrium” discussed in our manuscript specifically refers to the two-temperature (2T) state, where electrons and

ions are at different temperatures. We also concur with your point that, for systems truly far from equilibrium—where the electronic subsystem is highly non-thermal and the electron temperature is not well-defined—a kinetic model (such as the Boltzmann equation) is indeed required to accurately describe electron kinetics and transport.

In the context of ultrafast laser excitation of metals, it is well established that the lifetime of nonequilibrium (non-Fermi-Dirac) electrons is extremely short, typically less than 100 fs [Phys. Rev. Lett. 110, 135001 (2013)]. After this ultrafast thermalization, the electronic subsystem rapidly thermalizes, allowing the use of a well-defined electron temperature T_e and thus making a hydrodynamic (two-temperature) model for electrons appropriate and effective for describing the subsequent picosecond-scale atomic dynamics that are the focus of our study.

As you correctly pointed out, when the electron mean free path (MFP) is comparable to or exceeds the system size, the initial stage following laser excitation is dominated by ballistic electron transport—a kinetic effect. In our two-temperature model, the effect of ballistic transport is effectively incorporated by considering the resulting uniform energy deposition: ballistic electron motion quickly redistributes the deposited energy throughout the thin film, leading to a spatially homogeneous electron temperature profile [Phys. Rev. Lett. 108, 165001 (2012)]. This kinetic effect is thus equivalently included in our treatment of energy deposition, as described in the Methods section:

”Since both the thickness of W and Au nanofilm is comparable to the mean free path of excited electrons (~ 33 nm in W and ~ 100 nm in Au), the ballistic motion of the excited electrons leads to the fast (typically a few ~ 10 fs) redistribution of the deposited energy within the ballistic range. As a result, the uniform laser energy deposition is expected...”

Regarding the ”nonthermal effect”, the keypoint of this phenomenon lies in **the modification of the potential energy surface (PES) by hot electrons generated during laser excitation**. The electron temperature T_e serves as an effective descriptor for these hot-electron effects, as it characterizes the degree of electronic excitation induced by the laser [Phys. Rev. Lett. 96, 055503 (2006)][Phys Rev Lett 95, 085002 (2005)]. In our work, we specifically address the two-temperature ($T_e \gg T_i$) regime, where the elevated electron temperature fundamentally alters the PES and thus drives structural evolution. As shown in the following figure, the neural network model employed in this study explicitly incorporates electron temperature as a degree of freedom and is trained on extensive first-

principles data covering a wide range of electron temperature, lattice temperature, densities, and strains. This enables the model to accurately capture the changes in atomic forces and stresses induced by electronic excitation, and thus to describe nonthermal, hot-electron-driven structural transitions.

FIG. S1: Schematic diagram of electron temperature-dependent interatomic potential from [npj Comput. Mater. 9, 213 (2023)]. T_e is the electron temperature, regarding to the electron occupation distribution. The free energy A , force \mathbf{F} , virial Ξ , electronic entropy S_e , and electronic heat capacity C_e can be inferred through backpropagation algorithm.

Zeng, Q., Chen, B., Zhang, S. et al. Full-scale ab initio simulations of laser-driven atomistic dynamics. npj Comput Mater 9, 213 (2023). <https://doi.org/10.1038/s41524-023-01168-4>

We thank the reviewer again for raising this important point, which has helped us to clarify the scope and applicability of our modeling approach. We acknowledge that our approach does not explicitly resolve the full kinetic evolution of non-thermal electrons at the sub-100 fs timescale. However, for the picosecond-scale atomic and structural dynamics that are the focus of our work, the two-temperature model combined with an electron-temperature-dependent potential provides a physically justified and computationally tractable framework. We have clarified these modeling assumptions and their limitations in the 1st paragraph of revised *Method* section, and we agree that future work could benefit from more explicit kinetic modeling to further improve the description of non-equilibrium electron dynamics.

4. All of this is important because one needs to establish what the big discovery is that warrants a paper and one in this journal. This appears to be the "paradigm shift" due to hot

electrons exerting an additional pressure and the question of how this affects structural transformations. From what I can tell, this means using temperature-dependent input, a rather obvious idea. (In fact, it isn't clear why this would not be done for any system that has an electron temperature near or greater than the Fermi energy.) What I was expecting to see was a "smoking gun" related to the "vital role of the electronic pressure". Since electronic pressure is always present, what we are looking for is some comparison of the electron pressure relative to other terms spatially. The fact that there are more terms in a heterogeneous system is trivial - of course, spatial gradient terms become important when there are spatial gradients. That the electronic pressure is "vital" requires more effort to show. I always think of electronic pressure as being important: it drives many phenomena across a wide range of materials; for example, it is responsible for ambipolar diffusion in plasmas. The visualizations of the simulations can only partly explain this phenomena: back of the envelope, perhaps Thomas-Fermi based, to show the underlying physics in a simple way.

Reply:

Thank you very much for your thoughtful and constructive comments. We sincerely appreciate your careful reading and the important questions you raised regarding the novelty and physical significance of our work, especially concerning the role and demonstration of electronic pressure in laser-driven phase transitions.

First, we would like to clarify a key conceptual point: **in the context of ultrafast laser-induced phase transitions, the role of electronic pressure has not always been properly recognized or incorporated.** The emergence of significant electronic pressure is a direct consequence of hot electrons modifying the potential energy surface (PES) of the system. However, in most previous studies of laser-induced melting, the ground-state PES ($T_e = T_i$) is universally adopted, and the existence and relaxation dynamics of electronic pressure are completely neglected. As a result, the prevailing view in the literature has been to treat laser-driven processes as isochoric (constant volume) or purely thermal, with melting attributed solely to lattice heating through electron-phonon coupling [Phys. Rev. Lett. 121, 075002 (2018)][Matter Radiat. Extremes 7, 036901 (2022)][Nat Commun 12, 1638 (2021)].

On this basis, our work addresses two major challenges: (1) the lack of a practical and accurate approach to incorporate excited-state PES ($T_e \gg T_i$) in large-scale, real-time simulations, and (2) the absence of direct, quantitative evidence showing how electronic pressure

fundamentally alters the structural evolution pathway during laser-induced melting. Traditional molecular dynamics methods, based on ground-state PES, cannot capture nonthermal effects such as electronic pressure, leading to significant discrepancies with experimental observations—indeed, as shown by Arefev et al. [Sci. Adv. 8, eabo2621 (2022)], ultrafast electron diffraction (UED) experimental results cannot be reproduced within the ground-state framework without invoking unphysical parameters. On the other hand, first-principles approaches are limited to small, periodic systems ($10^1 - 10^2$ atoms) and cannot resolve the relaxation dynamics of electronic pressure in realistic, free-surface nanofilms ($> 10^5$ atoms). As summarized in the 3rd paragraph of *Introduction* section,

”Theoretically, the laser-driven process presents dual requirements that existing methodologies struggle to reconcile: *ab initio* accuracy to capture the hot-electron-modified PES governing p_e , and atomic-scale resolution across experimentally relevant thickness (tens of nanometers) to track stress wave propagation and phase transition. *Ab initio* accuracy and large-scale simulation size significantly constrains the efficacy of existing methodologies, such as classical molecular dynamics combined with two-temperature model or real-time time dependent density functional theory, thereby hindering a thorough exploration into the real-time response of laser-excited material without any prior constraints.”

In our main text, we directly address these gaps by providing a clear and explicit comparison between ground-state and excited-state PES in laser-induced melting. When the ground-state PES is used, the system exhibits a homogeneous, thermally driven melting pathway (Fig.1). In stark contrast, when the excited-state PES (including hot electron effects) is considered, the relaxation of electronic pressure drives an surface-initiated heterogeneous melting process in W nanofilms, but the melting speed (2500 m s^{-1}) is tenfold greater than that of conventional heterogeneous melting process ($\sim 10^2 \text{ m s}^{-1}$). These two scenarios represent fundamentally different mechanisms of structural evolution. Furthermore, our thermodynamic pathway analysis (Fig.2, Fig.S2, Fig.S3) reveals that electronic pressure leads to rapid, nonthermal expansion and dramatic density reduction—features that cannot be captured by ground-state models. We also show that the time-resolved X-ray diffraction (XRD) signatures predicted by our simulations can distinguish this nonthermal expansion (driven by electronic pressure) from conventional thermal expansion (driven by thermoelastic stress), providing an experimentally accessible benchmark for our findings (Fig.4 and Fig.5).

Fig. 1: **Diverging melting dynamics in laser-excited tungsten.** Under absorbed laser fluence of 120 mJ cm^{-2} , (a) Conventional TTM-MD prediction with ground-state PES $A(\mathcal{R})$ showing homogeneous melting governed by electron-phonon coupling. (b) Our TTM-DPMD results with laser-excited PES $A(\mathcal{R}, T_e)$ revealing electronic pressure-driven heterogeneous melting. The local structures are identified by the polyhedral template matching (PTM) method, and polyhedral surface meshes around FCC-type (green) and amorphous-type (gray) particles are constructed to highlight the heterogeneity in lattice symmetry. Red arrows indicate the propagation direction of phase transition (solid-liquid or BCC-FCC) interfaces.

FIG. S3: (a) The temporal evolution of surface velocity (b)(c) thermodynamic pathway of selected region in laser-excited W nanofilm under laser fluence of 120 mJ cm^{-2} in the first 7 ps, where simulation results with ground-state PES are presented for comparison.

Fig. 5: Temporal evolution of static structure factor $S(q)$ of laser-excited W nanofilm at different laser fluence, estimated from TTM-DPMD trajectories with (a) ground-state PES (b) laser-excited PES. The positions of new diffraction peak are marked by black dashed line to highlight the difference between thermal and nonthermal expansion dynamics.

In summary, the key discovery of our work is the identification and quantitative demonstration of a **universal, hot-electron-mediated pathway for ultrafast heterogeneous melting and structural transformation in metals**, which is fundamentally governed by the relaxation dynamics of electronic pressure. This is not simply a matter of including a temperature-dependent input, but rather of demonstrating—through large-scale, atomistic, and *ab initio* accurate simulations—how the build-up and subsequent relaxation of electronic pressure leads to new, nonthermal pathways for phase transitions that are qualitatively distinct from conventional thermal or instability-driven mechanisms. This challenges the long-standing isochoric assumption in laser-matter interaction studies and provides new insights into the interpretation of ultrafast experiments and the design of laser-based material processing. We hope that these clarifications address your concerns and highlight the novelty and significance of our results.

Thank you again for your valuable feedback, which has greatly improved the clarity and

impact of our work.

5. Because of the heterogeneity of the system, and the large electron MFP, we expect the free energy A to have time-evolving spatial gradients. How was the ETD-DP trained on various temperatures densities and each of their gradients? Or, was A trained on homogeneous systems and used in a "slowly varying" approximation here by using a local density in A ?

Reply:

Thank you for this insightful question regarding the training and applicability of the ETD-DP (electron-temperature-dependent deep potential) model in the context of spatial heterogeneity and large electron mean free path.

First, due to the ballistic transport of excited electrons immediately after ultrafast laser irradiation, the electron temperature in our simulated thin films is rapidly homogenized, as discussed in the *Method* section and *Response to Question 3*. As a result, the main source of spatial heterogeneity in our simulations arises from the ionic subsystem—namely, variations in ion temperature and local density—rather than from the electron temperature field itself.

The ETD-DP model is constructed using a local (atom-centered) approximation, in which the total free energy is expressed as a sum over atomic contributions, $A = \sum_i A_i$, to ensure the extensivity of the system energy. Each atomic free energy A_i is a function of its local atomic environment and the local electron temperature. The training dataset systematically samples a wide range of ion temperatures (100–4000 K), electron temperatures (100–25,000 K), and densities (via different lattice constants and uniaxial strains along multiple crystallographic directions), as described in the *Methods* section. Importantly, the training data include both fully relaxed and partially relaxed configurations under various temperature and pressure conditions, covering not only equilibrium structures but also configurations with lattice distortions, thermal fluctuations, and solid-liquid interfaces. This diversity ensures that the model can accurately interpolate to a broad range of local environments encountered in spatially heterogeneous systems. In practice, the model operates under a "slowly varying" approximation, where the local atomic environment is used to predict the free energy and forces for each atom, which is well justified by the short-range nature of atomic interactions in condensed matter and has been widely validated in the machine-learning PES literature [Behler & Parrinello, *Phys. Rev. Lett.* 98, 146401 (2007); Zhang et al., *Phys. Rev. Lett.* 120, 143001 (2018)].

In summary, the ETD-DP model is trained on a wide range of local environments, and its local, extensible form allows it to capture the essential physics of spatial heterogeneity in our simulations. The main spatial gradients in our system arise from the ionic subsystem, which are well described by the model. We have clarified these points in the revised manuscript and thank the reviewer for raising this important aspect.

6. In summary, I do not recommend publication of this manuscript. Apart from the writing and unclear descriptions, it is not clear that a significant problem has been solved. The manuscript reads as series of toy experiments with simple observations based on extracted observables. The experiments don't entirely focus on the main theme and the observables don't prove the point the authors appear to be interested in.

Reply:

We sincerely thank the reviewer for the thoughtful feedback and the opportunity to clarify the scientific focus and significance of our work.

We agree that clearly articulating the central problem and its broader impact is essential. The microscopic mechanism of laser-induced melting under two-temperature (2T) state is indeed a central and actively debated topic in ultrafast materials science and high-energy-density physics [Physics Reports 1089, 1 (2024); Science 360, 1451 (2018); Sci. Adv. 10, eadh5272 (2024)]. In particular, the role of hot-electron effects—especially electronic pressure—in driving phase transitions remains unresolved, despite extensive experimental and theoretical efforts in recent years.

Our study directly addresses this knowledge gap by, providing large-scale atomistic simulations that incorporate excited-state PES and explicitly track the build-up and relaxation of electronic pressure in realistic nanofilms. This approach allows us to reveal a nonthermal, electronic-pressure-driven melting pathway that is fundamentally different from conventional thermal melting.

To ensure our work is clearly focused and experimentally relevant, we have systematically organized the manuscript around the following key question (as stated in the *Introduction* section on the manuscript): "How does the accumulation of electronic pressure (during $\sim 10^2$ fs) and its relaxation (on $\sim 10^1$ ps timescale) affect structural transformations?" We address this by (1) Comparing ground-state and excited-state PES simulations to isolate the effect of electronic pressure. (2) Quantitatively analyzing thermodynamic pathways, structural

evolution, and the propagation of melting fronts under different laser fluences and sample thicknesses. (3) Providing direct predictions for experimentally accessible observables, such as time-resolved X-ray diffraction (XRD) peak splitting.

Contrary to the impression of “toy experiments,” as commented by Reviewer 1 and Reviewer 2, our approach is designed to bridge atomistic theory and experiment, and to provide new physical insights that are highly relevant to ongoing research in ultrafast laser-matter interactions. The TTM-DPMD framework we employ has been systematically validated: it accurately reproduces the thermophysical and acoustic properties of tungsten (W) under both equilibrium and non-equilibrium conditions, as demonstrated in our previous work [npj Comput. Mater. 9, 213 (2023)]. Furthermore, the laser-driven dynamics simulated by this approach are in good agreement with ultrafast electron diffraction (UED) experimental results for W nanofilms reported by Mo et al. [Sci. Adv. 5, eaaw0392 (2019)]. We believe that this combination of first-principles accuracy, large-scale simulation capability, and direct experimental relevance ensures the robustness and significance of our findings. We hope that this revised and clarified presentation makes the significance and main theme of our work more apparent.

Thank you again for your constructive comments, which have helped us to further strengthen the manuscript.

Thank you all again

Sincerely yours,

Qiyu Zeng, Xiaoxiang Yu, Bo Chen, Shen Zhang, Kaiguo Chen, Dongdong Kang, Jiayu Dai

Response

(Dated: September 9, 2025)

Manuscript ID: NCOMMS-25-13275

Title: Ultrafast Heterogeneous Melting of Metals under Extreme Non-equilibrium States

Dear Editors and Reviewers,

We are grateful to the reviewers for their time and insightful comments, which have significantly improved the quality of our manuscript. We are pleased that both reviewers support publication. Below is our point-by-point response to the final comments.

REPLY TO REVIEWER #2

Reviewer #2 (Remarks to the Author):

The authors have addressed all my comments correctly, and I am satisfied with the revisions. However, I'd only remark that Figure 2 would be better displayed if the panels are oriented vertically instead of horizontally. For instance, like in the scheme below.

A.B.

C.D.

E.F.

Moreover, the panels currently in 'C' would need to be labelled and be at similar size as panels A and B. At the moment, panels in C look very small, and we barely can see the details. After that, the paper can be accepted for publication.

Reply:

We sincerely thank the reviewer for this excellent suggestion and for their endorsement of our work. We have completely redesigned Figure 2 according to the reviewer's recommendation to improve its clarity and visual impact. The specific changes include:

1. The figure panels are now arranged in a clear vertical orientation (AB, CD, EF) as suggested.
2. The original panel C has been split into four distinct, larger panels. This allows each sub-figure to be displayed at a much larger size.

We believe the new Figure 2 is a substantial improvement and effectively addresses the reviewer's concern. It is now included in the revised manuscript.

FIG. 2: Extreme heterogeneity in laser-excited W. (a)(b) Depth-dependent thermodynamic pathways in tungsten nanofilm under laser fluence of 120 mJ cm^{-2} . Colored circles show thermodynamic states at 0.5 ps intervals, with arrows indicating evolution trajectories. Red stars mark three representative thermodynamic states of surface region ($z = 1.8 \text{ nm}$) corresponding to: initial electronic pressure buildup ($t = 0.5 \text{ ps}$), complete pressure release ($t = 1.5 \text{ ps}$), and surface melting ($t = 3 \text{ ps}$). Blue stars denote simultaneous thermodynamic states of interior region ($z = 14.2 \text{ nm}$) for comparison. The calculated equilibrium isochore (gray solid line) and melting curve (black solid line) are also presented for comparison. (c)(e) Radial distribution functions $g(r)$ and (d)(f) coordination numbers (CN) for surface and interior regions at selected times, where radial distribution of CN are obtained by integration of $g(r)$.

REPLY TO REVIEWER #3

Reviewer #3 (Remarks to the Author):

The authors have done a fine job of improving the manuscript and I support publication.

Reply:

We thank the reviewer for their positive feedback and for supporting the publication of our manuscript.

Thank you all again

Sincerely yours,

Qiyu Zeng, Xiaoxiang Yu, Bo Chen, Shen Zhang, Kaiguo Chen, Dongdong Kang, Jiayu Dai